# AgentTailor: A Semantic-Aware LLM-Based Multi-Agent System with Actor-Critic Structure

**Peiting Yang** [* 1] **Jiahao Shi** [* 1] **Caiyi Xu** [* 1] **Ming Liu** [2] **Yanxia Wu** [1] **Rongsheng Li** [1]

## Abstract

Large Language Model (LLM)-based multi-agent systems often suffer from high communication cost due to redundant interactions, as existing methods optimize communication structures without explicitly measuring whether exchanged messages contribute to the final decision. To better utilize the semantic information in the execution stage to further optimize the structure of multi-agent systems and reduce token costs, we propose **AgentTailor**, a cost-aware framework that evaluates the semantic contribution of communication edges via an edge judgment mechanism, and employs an **Edge Prediction Network (EPN)** to estimate edge utilities through virtual execution without invoking LLMs. Experiments show that AgentTailor achieves the best average accuracy (91.36%) on six datasets of diverse fields, while reducing total tokens by 21.2%–61.6%. Our work demonstrates that explicitly modeling semantic edge contribution is crucial for scalable and efficient multi-agent systems, providing a principled approach to communication optimization that goes beyond structural heuristics. The code is available at `https://github.com/Pt3Y/AgentTailor`.

## 1. Introduction

Recent advances in large language models (LLMs) have catalyzed rapid progress in multi-agent systems (MAS), enabling groups of agents to collaboratively solve complex tasks such as reasoning, planning, and decision-making (Wu et al., 2023; Zhuge et al., 2024; Hong et al., 2024; Qian et al., 2024b). By decomposing problems and allowing agents to exchange intermediate information, LLM-based MAS have achieved impressive results across a wide range of benchmarks (Chan et al., 2023; Chen et al., 2023). Despite these successes, the efficiency and scalability of multi-agent communication remain critical challenges (Lyu et al., 2024; Zhang et al., 2024; Wang et al., 2025b).

A central issue lies in the communication structure among agents. In many existing systems, agents are either fully connected or communicate according to fixed or heuristically designed topologies (Wu et al., 2023; Zhuge et al., 2024). Such designs inevitably introduce substantial redundancy, as not all communication edges contribute equally to the final outcome (Zhang et al., 2024; Wang et al., 2025b). To address this, recent work has explored learning sparse communication structures(Leong et al., 2025; Li et al., 2025a; Hanchen Jiang et al., 2025), often by assigning activation probabilities to edges or pruning edges based on structural importance (Zhang et al., 2024; Wang et al., 2025b). However, these approaches typically focus on *whether* an edge is activated, rather than *how* the information transmitted by that edge is actually used during execution. In practice, an agent may receive numerous messages, but only a subset of them meaningfully influences the final decision (Zhang et al., 2024; Wang et al., 2025b). Treating all activated edges as equally important obscures this distinction and leads to inaccurate credit assignment (Li et al., 2025c). As a result, structure learning and pruning methods may preserve redundant or even harmful communication while discarding early but critical reasoning paths (Zhang et al., 2024; Fu et al., 2024). This limitation becomes particularly pronounced in LLM-based systems, where communication cost directly translates into token consumption and latency (Zhang et al., 2024; Fu et al., 2024).

To address this issue, we argue that effective communication optimization in MAS should be grounded in semantic contribution that can be **explicitly measured**, rather than mere activation. Specifically, at the edge level, an edge should be considered important if the information it conveys is substantively adopted in the final decision of the current interaction round. Based on this criterion, we propose **AgentTailor**, a framework that explicitly measures and exploits semantic edge contribution to guide communication structure learning.

---

[*]Equal contribution [1]College of Computer Science and Technology, Harbin Engineering University, Harbin, China [2]School of Information Technology, Deakin University, Burwood, Australia. Correspondence to: Rongsheng Li <dasheng@hrbeu.edu.cn>.

*Proceedings of the $43^{rd}$ International Conference on Machine Learning*, Seoul, South Korea. PMLR 306, 2026. Copyright 2026 by the author(s).

AgentTailor introduces an **edge contribution evaluation framework** that quantifies the semantic alignment between the message transmitted by an edge and the final aggregated output of the system, reflecting the extent to which each edge's information contributes to the final decision. Within this framework, directly comparing an edge's transmitted message to the final output can introduce a systematic distortion we term **positional bias**: edges occurring later in the communication sequence naturally tend to show higher similarity to the final output, regardless of their true informational value. To correct for this, we define a **relative contribution** measure that normalizes each edge's utility against the average contribution of its source node's incoming edges, enabling AgentTailor to identify edges that provide genuine incremental value rather than merely transmitting redundant information. To make the framework highly scalable and lightweight while preserving accurate semantic-level credit assignment, we construct an **Edge Prediction Network (EPN)** that approximates edge utilities and relative contribution measures through **virtual execution**. This decouples structure evaluation from costly LLM calls and makes the framework potentially scalable to large-scale communication settings.

We evaluate the effectiveness of this framework through extensive experiments, demonstrating how the edge-level utility and relative contribution measures correlate with improved system performance (details in Section 4). Under this framework, AgentTailor achieves the best average accuracy of 91.36% across six datasets of diverse fields, with a prominent decrease of token consumption of 21.2%–61.6% compared with AgentDropout under same configuration. Moreover, robustness experiments have shown AgentTailor's stable performance under adversarially misleading agents.

**Contributions.** Our main contributions are summarized as follows:

- We identify a key limitation of existing multi-agent communication learning methods: the lack of semantic-level credit assignment for communication edges.

- We propose an edge judgment mechanism that explicitly measures how much each communication edge's information is adopted in the final decision, together with a relative contribution normalization to eliminate positional bias.

- We introduce an Edge Prediction Network that enables virtual execution for efficient structure learning without invoking LLMs.

- We conduct extensive experiments to provide points of comparison and references for method performance,

serving as a resource for future research on multi-agent communication.

## 2. Related Work

Multi-agent systems based on large language models (LLMs) have shown strong capabilities in complex reasoning, decision-making, and collaborative problem solving (Guo et al., 2024; Ni & Buehler, 2024; Chen et al., 2023; Qian et al., 2024b). Representative frameworks such as AutoGen (Wu et al., 2023), GPTSwarm (Zhuge et al., 2024), and MetaGPT (Hong et al., 2024) organize multiple agents into structured interaction pipelines, enabling collective intelligence beyond single-agent performance. Other systems, including ChatEval (Chan et al., 2023), AgentVerse (Chen et al., 2023), and MacNet (Qian et al., 2024b), explore advanced multi-agent coordination with graph-based communication and emergent behaviors (Li et al., 2025b; Furuya & Kitagawa, 2025).From a communication-structure perspective, existing systems adopt a variety of predefined topologies. These include non-interactive settings (Zhang et al., 2023; Du et al., 2023), sequential chain structures (Qian et al., 2024a; Hong et al., 2024; Holt et al., 2023), centralized star architectures(Wu et al., 2023; Yan et al., 2024; Zhou et al., 2023), hierarchical tree organizations(Ishibashi & Nishimura, 2024), and graph-based schemes with dense or partial connectivity (Qian et al., 2024b; Zhuge et al., 2024). While useful as inductive biases, these structures are typically manually designed or heuristic, introducing token overhead and limiting scalability (Hanchen Jiang et al., 2025; Leong et al., 2025; Syed et al., 2025). Naive scaling may further reduce efficiency and performance (Wang et al., 2025a; Furuya & Kitagawa, 2025).To improve communication efficiency, several methods learn or prune agent communication patterns. AgentPrune (Zhang et al., 2024) performs one-shot pruning on a spatio-temporal communication graph, while AgentDropout (Wang et al., 2025b) dynamically removes redundant agents and links (Wang et al., 2025a). G-Designer (Zhang et al., 2025) advocates generating graphs conditioned on the given problem, yet it similarly fails to leverage edge-level semantics during graph construction. Other graph- or learning-based methods optimize connectivity, edge selection, and prompt/inference structures (Syed et al., 2025; Wang et al., 2025a; Furuya & Kitagawa, 2025).Another line of research targets computational and economic efficiency(Chen et al., 2025; Yang et al., 2025a; Pitre et al., 2025; Yang et al., 2025c), including token-efficient prompting, economical agent coordination, and lightweight communication protocols, balancing accuracy and budget (Yang et al., 2025b; Lackinger et al., 2025). Nevertheless, robustness and stability remain underexplored, as learned topologies can be sensitive to agent configurations or training dynamics (Li et al., 2025b; Hanchen Jiang et al., 2025; Furuya & Kitagawa, 2025), underscoring the

importance of cost-aware, semantically informed communication optimization for practical deployment (Furuya & Kitagawa, 2025).

## 3. Methodology

### 3.1. Overview

As Figure 1 shows, AgentTailor integrates three core modules: an Actor that parameterizes spatial and temporal edge probabilities for the communication graph, a semantic Edge Judge that measures the contribution of each message to the final decision, and a Critic including an Edge Prediction Network (EPN) that learns to predict edge utilities without additional LLM calls.

The Edge Judge provides fine-grained semantic supervision for executed edges, while the EPN distills this supervision into a compact predictor that can be reused across queries. Together, they allow the Actor to focus probability mass on semantically useful edges rather than relying only on structural heuristics.

In the *training stage*, the Edge Judge computes semantic supervision by comparing each current message with the final aggregated message of the same round, and the Critic (EPN) learns to predict these edge-level signals from the query, role, and history representations. After the Critic has been warmed up to provide approximate edge-utility estimates that are informative enough for ranking during virtual execution—but are not assumed to be precisely calibrated—we insert $K_v$ virtual executions (guided by EPN, without LLM calls) before each real execution to refine the communication graph at low token cost. In the *testing stage* (bottom of Figure 1), we first perform per-query virtual tailoring using the learned Critic for $K_v^{(3)}$ steps (see Table 7), prune low-probability edges, and then run a final real execution to produce the answer.

This design yields a semantically informed, token-efficient communication policy that adapts the graph per task while minimizing expensive LLM calls, and cleanly separates costly supervision collection from cheaper policy refinement and test-time adaptation.

### 3.2. The Logic of Edge Judgment

In multi-agent systems (MAS), not all edges contribute equally to the final answers. Many reinforcement learning-based methods consider the contribution of edges only through their activation probabilities, which **disregards the actual usage of messages during execution**. AgentTailor introduces an edge-level scoring mechanism that assigns a utility score to each executed edge, directly reflecting how much the information conveyed by that edge is adopted in the **final aggregated message of the current round**, which

can guide structure learning and pruning more accurately.

**Definition of Edge Utility.** For an edge $e_{ij}$ (from node $i$ to node $j$), we extract the textual message passed from agent $i$ to agent $j$ and encode it using a pretrained sentence transformer encoder (specifically, `all-MiniLM-L6-v2`) into a fixed-dimensional vector $\mathbf{h}_{ij} = \text{Enc}(\text{message}_{ij})$, where Enc denotes the sentence encoder. Simultaneously, we extract **the final aggregated message of the current round**, which corresponds to the input to the final decision node (i.e., the concatenation of all agents' outputs that are used to produce the final answer in this round), and encode it using the same encoder as $\mathbf{h}_{\text{final}} = \text{Enc}(\text{final\_aggregated\_message})$. The utility of the edge $e_{ij}$ is then defined as the cosine similarity between these two embeddings:

$$U_{ij} = \cos(\mathbf{h}_{ij}, \mathbf{h}_{\text{final}}) = \frac{\mathbf{h}_{ij} \cdot \mathbf{h}_{\text{final}}}{\|\mathbf{h}_{ij}\| \cdot \|\mathbf{h}_{\text{final}}\|}. \quad (1)$$

This utility measures how strongly the information conveyed by this edge is adopted in the final output message of the current round. $U_{ij}$ close to 1 indicates the edge's information is heavily adopted in this round, while values close to 0 or even negative signs for this information were considered as harmful or useless information in this round.

**Normalization via Relative Contribution.** However, this utility will be deeply related with the position of this edge. An edge closer to the final node would get a higher utility more easily. To eliminate the effect of position and normalize across a round, we compare each outgoing edge with the average contribution of all incoming edges to its source node. For edge $e_{ij}$ (from node $i$ to node $j$), we define its **relative contribution**, denoted as $\Delta_{ij}$, as:

$$\bar{U}_i = \frac{1}{|\mathcal{E}_i|} \sum_{e_{ki} \in \mathcal{E}_i} U_{ki},$$
$$\Delta_{ij} = U_{ij} - \bar{U}_i, \quad (2)$$

where $\mathcal{E}_i = \{e_{ki} \mid e_{ki} \text{ is an incoming edge to node } i\}$ denotes the set of all edges pointing into the source node $i$, $\bar{U}_i$ is the average utility of these incoming edges, and $\Delta_{ij}$ measures how much more (if $\Delta_{ij} > 0$) or less (if $\Delta_{ij} < 0$) the outgoing edge $e_{ij}$ contributes compared to the average contribution of edges that provide input to its source node $i$.

By explicitly computing how much edge information is adopted in the final decision, AgentTailor can more accurately identify which edges truly contribute to task completion and which are redundant, enabling more effective structure pruning while maintaining or improving performance. These edge-level signals are then used to supervise the Critic and to construct rewards for the Actor.

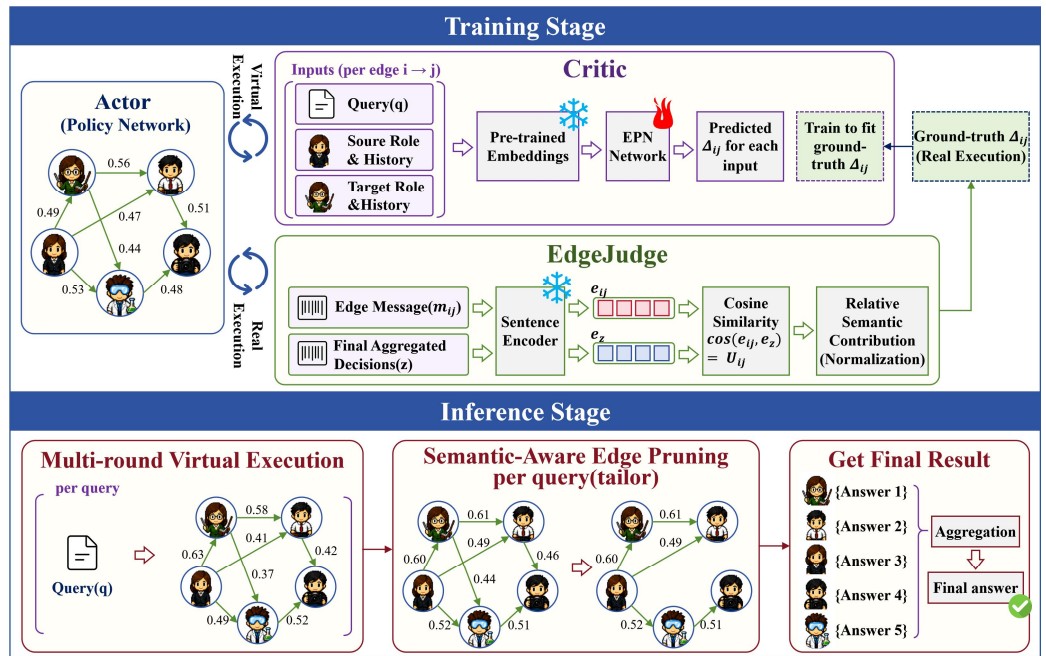

*Figure 1.* Overview of the proposed AgentTailor framework and training stage. AgentTailor's framework consists of an Actor to sample graph structure, a semantic Edge Judge to evaluate the contribution of edge, and a Critic with an Edge Prediction Network (EPN) that predicts the contribution and supports virtual execution and pruning.

### 3.3. Actor-Critic Structure

**Reward Computation.** After computing relative contributions $\Delta_{ij}$ via the Edge Judge, we normalize them to construct rewards:

$$r_{ij} = \frac{\Delta_{ij}}{|\sum_{e_{kl} \in \mathcal{E}_t} \Delta_{kl}| + \epsilon}, \quad (3)$$

where $\mathcal{E}_t$ denotes edges executed in round $t$, and $\epsilon > 0$ prevents division by zero. This normalization preserves the sign of $\Delta_{ij}$ while scaling rewards relative to the total contribution magnitude in each round.

**Policy Gradient Loss.** The Actor is updated using REINFORCE-style policy gradient. The loss function combines reward-weighted log probabilities with sparsity regularization:

$$\mathcal{L}_{\text{real}} = -\frac{1}{|\mathcal{E}_{\text{executed}}|} \sum_{e_{ij} \in \mathcal{E}_{\text{executed}}} \log(p_{ij}) \cdot r_{ij} +$$

$$\lambda_{\text{sparsity}} \cdot \left( \frac{1}{|\theta_S|} \sum_{l_{ij} \in \theta_S} |l_{ij}| + \frac{1}{|\theta_T|} \sum_{l_{ij} \in \theta_T} |l_{ij}| \right),$$

$$(4)$$

where $\mathcal{E}_{\text{executed}}$ is the set of executed edges, and $\lambda_{\text{sparsity}}$ controls sparsity regularization strength.

**Gradient Update.** The gradient with respect to logit $l_{ij}$ is:

$$\frac{\partial \mathcal{L}_{\text{real}}}{\partial l_{ij}} = -r_{ij} \cdot (1 - p_{ij}) + \lambda_{\text{sparsity}} \cdot \text{sign}(l_{ij}), \quad (5)$$

where $(1 - p_{ij})$ is the gradient of $\log(\sigma(l_{ij}))$. This ensures positive rewards increase $l_{ij}$ (and thus $p_{ij}$), while negative rewards decrease them. The Actor parameters are updated via:

$$\theta_S \leftarrow \theta_S - \alpha_{\text{actor}} \nabla_{\theta_S} \mathcal{L}_{\text{real}}, \quad \theta_T \leftarrow \theta_T - \alpha_{\text{actor}} \nabla_{\theta_T} \mathcal{L}_{\text{real}},$$

$$(6)$$

where $\alpha_{\text{actor}}$ is the Actor learning rate.

**Relation to classical Actor–Critic.** In classical actor–critic methods, the critic is a value function that estimates long-term returns and enters the policy gradient as a baseline or advantage to reduce variance. In contrast, our Critic (EPN) regresses edge-level semantic contribution signals $\Delta_{ij}$ from the Edge Judge and serves as a semantic reward surrogate for real and virtual execution, rather than a traditional state/action value baseline.

### 3.4. EPN: Edge Prediction Network

To capture the patterns of edge utility and provide prediction, we introduce the Edge Prediction Network (EPN), a neural network that predicts the relative contribution $\Delta_{ij}$ from historical messages, role and tasks without calling LLM.

This neural network serves as the core component of Critic, allowing our AgentTailor to execute virtually, which means more chances to train and tailor the graph based on task without increasing token cost.

**Network Architecture.** The input of EPN contains 5 components: (1) the description of source node, including its role and prompt; (2) the historical outputs of source node; (3) the original task query; (4) the description of target node, including its role; (5) the historical outputs of target node. Each of these messages is encoded into a 384-dimensional vector by a pretrained sentence transformer (e.g.,`all-MiniLM-L6-v2`), and the five vectors are concatenated to form the EPN input. We denote by $\mathbf{x}_{ij}$ the concatenation of the encoded source-node description and history, the task query, and the target-node description and history, which serves as the input to the EPN. The EPN consists of a multi-layer perceptron (MLP) followed by a residual block. The formal computation of EPN is as follows:

$$
\begin{aligned}
\mathbf{h}^{(0)} &= \mathbf{x}_{ij}, \\
\mathbf{h}^{(\ell)} &= \text{ReLU}(\mathbf{W}_\ell \mathbf{h}^{(\ell-1)} + \mathbf{b}_\ell), \quad \ell = 1, \ldots, L-1,
\end{aligned}
\tag{7}
$$

where $L$ is the number of hidden layers, $\mathbf{W}_\ell$ and $\mathbf{b}_\ell$ are the weight matrix and bias vector of the $\ell$-th layer, and $\mathbf{h} = \mathbf{h}^{(L-1)}$ is the final hidden representation. The hidden representation $\mathbf{h}$ is then enhanced through a residual connection:

$$
\mathbf{h}' = \mathbf{h} + \text{Dropout}(\text{Linear}(\text{GELU}(\text{Linear}(\mathbf{h})))), \tag{8}
$$

where the residual block consists of two linear layers with GELU activation. Finally, the predicted relative contribution is computed as:

$$
\hat{\Delta}_{ij} = \tanh(T \cdot (\mathbf{W}_o \mathbf{h}' + b_o)), \tag{9}
$$

where $\mathbf{W}_o$ and $b_o$ are the output layer parameters, and $T > 0$ is a temperature hyperparameter that controls the output range. The activation of $\tanh$ ensures that the predictions are bounded in $(-1, 1)$, while the temperature scaling allows fine-grained control over the prediction sensitivity.

**Training Objective.** The EPN is trained to minimize the mean squared error between predicted and actual relative contributions:

$$
\mathcal{L}_{\text{EPN}} = \frac{1}{|\mathcal{E}_{\text{executed}}|} \sum_{e_{ij} \in \mathcal{E}_{\text{executed}}} (\hat{\Delta}_{ij} - \Delta_{ij})^2, \tag{10}
$$

where $\mathcal{E}_{\text{executed}}$ represents the set of edges that were executed or selected, and $\Delta_{ij}$ is the ground-truth relative contribution computed using the Edge Judge (as defined in Section 3.1). The EPN is trained jointly with the Actor at first.

**Locking Mechanism.** To reduce computing costs, we adopt a mechanism of locking. When the loss of EPN falls below the pre-set threshold $\tau_{\text{lock}}$, we freeze the weights of EPN and encoder. The locked network serves as a virtual environment, could update the logits of our graph using only tasks without expensive LLM calls.

### 3.5. Virtual Execution

Virtual execution leverages the learned Critic (EPN) to approximate edge-level semantic rewards without additional LLM calls. For a given task query $q$ from a certain dataset, the process proceeds as follows:

**Edge Sampling:** Based on the current edge probability $p_{ij} = \sigma(l_{ij})$, we sample a set of edges $\mathcal{E}_{\text{sampled}}$ under the DAG constraint.

**Delta Prediction:** For each sampled edge $e_{ij} \in \mathcal{E}_{\text{sampled}}$, we construct the input feature vector for EPN from the current Actor state (node information and task query), and use EPN to predict its relative contribution:

$$
\hat{\Delta}_{ij} = \text{EPN}(\mathbf{x}_{ij}). \tag{11}
$$

**Reward Computation:** The predicted relative contributions $\hat{\Delta}_{ij} = \text{EPN}(\mathbf{x}_{ij})$ are normalized across rounds to compute fitted rewards:

$$
\hat{r}_{ij} = \frac{\hat{\Delta}_{ij}}{|\sum_{e_{kl} \in \mathcal{E}_t} \hat{\Delta}_{kl}| + \epsilon} = \frac{\text{EPN}(\mathbf{x}_{ij})}{|\sum_{e_{kl} \in \mathcal{E}_t} \text{EPN}(\mathbf{x}_{kl})| + \epsilon}, \tag{12}
$$

where $\mathcal{E}_t$ denotes the set of sampled edges in round $t$.

**Actor Update:** The Actor is updated using policy gradient methods with rewards computed from EPN-fitted predictions, along with sparsity loss:

$$
\begin{aligned}
\mathcal{L}_{\text{virtual}} = &-\frac{1}{|\mathcal{E}_{\text{sampled}}|} \sum_{e_{ij} \in \mathcal{E}_{\text{sampled}}} \log(p_{ij}) \cdot \hat{r}_{ij} + \\
&\lambda_{\text{sparsity}} \cdot \left( \frac{1}{|\theta_S|} \sum_{l_{ij} \in \theta_S} |l_{ij}| + \frac{1}{|\theta_T|} \sum_{l_{ij} \in \theta_T} |l_{ij}| \right),
\end{aligned}
\tag{13}
$$

where $\hat{r}_{ij}$ is the fitted reward computed from EPN-predicted relative contributions $\hat{\Delta}_{ij}$ via normalization, enabling Actor updates without real LLM execution.

**Adaptive Learning Rate.** EPN's ability to predict relative contribution has deep relation to the accuracy: the better the current graph performs, the more EPN can collect features from it. Therefore, we set the learning rate for virtual execution adaptively:

$$
\alpha_{\text{virtual}} = \max(\lambda_2 \cdot \text{acc}_{\text{current}}, \alpha_{\text{min}}), \tag{14}
$$

$\lambda_2$ is a hyperparameter and $\alpha_{\text{min}}$ is a minimum learning rate threshold. This adaptive scaling ensures that virtual updates

are more conservative when the model's performance is uncertain, and more aggressive when the model has learned reliable patterns.

**Integration with Real Execution.** In Stage 2, for each training sample we first perform $K_v$ virtual execution steps, followed by one real execution step. Virtual executions provide cheap policy exploration guided by EPN-predicted rewards, while real executions use actual $\Delta_{ij}$ from edge utilities to update both Actor and Critic. This hybrid schedule reduces token consumption while maintaining learning accuracy.

**Problem-Specific Graph Tailoring.** During evaluation, we use virtual execution to adapt the communication graph for each test query $q$: (1) reset the graph to the Stage-2 snapshot to ensure reproducibility, (2) perform $K_v^{(3)}$ virtual execution steps using the loss in Equation (13) to refine edge probabilities for $q$, (3) prune the bottom $p\%$ of edges by probability:

$$\mathcal{E}_{\text{prune}} = \text{BottomK}(\{p_{ij} : e_{ij} \in \mathcal{E}_{\text{active}}\}, \lceil |\mathcal{E}_{\text{active}}| \cdot p \rceil),$$
$$m_{ij} = 0 \quad \forall e_{ij} \in \mathcal{E}_{\text{prune}}, \tag{15}$$

where $\mathcal{E}_{\text{active}}$ denotes active edges after virtual refinement, and $m_{ij} \in \{0, 1\}$ is the mask indicating whether edge $e_{ij}$ is active. Finally, we execute the pruned graph with real LLM calls, so that EPN-guided virtual execution produces query-specific sparse topologies while keeping inference token cost low.

# 4. Experiments

We evaluate AgentTailor through a series of experiments, focusing on performance and token efficiency, robustness against attacks, and hyperparameter sensitivity and ablation studies. In our experiments, we focus on several central problems: (Q1) performance and token efficiency, (Q2) robustness against attacks, (Q3) hyperparameter sensitivity tests and ablations.

## 4.1. Experimental Setup

### 4.1.1. DATASETS AND TASKS

Six datasets are adopted to evaluate the ability of Agent-Tailor: **MMLU**(Hendrycks et al., 2020), a dataset on general reasoning across diverse subjects such as humanities, social sciences. **GSM8K**(Cobbe et al., 2021), a dataset of grade-school math word problems. **AQuA**(Ling et al., 2017), a dataset of algebraic multiple-choice reasoning problems. **HumanEval**(Chen, 2021), a code generation dataset. **SVAMP**(Patel et al., 2021), a dataset of arithmetic word problems requiring multi-step reasoning. **MultiArith**(Roy & Roth, 2015), a dataset of multi-step arithmetic word problems.

These datasets cover three main directions of LLM-based agent capabilities: general reasoning (MMLU), mathematical reasoning (GSM8K, SVAMP, MultiArith, AQuA), and code generation (HumanEval). Using these datasets, the ability of AgentTailor could be evaluated from diverse dimensions and application scenarios.

### 4.1.2. BASELINES

AgentTailor was compared with several other methods: 1.**Static structure** as **AutoGen**(Wu et al., 2023), a static chain-based multi-agent communication framework using history-passing conversation mechanism. 2.**Spatial optimizable structure** as **GPTSwarm**(Zhuge et al., 2024), a computational graph-based multi-agent framework with only spatial-edge connectivity optimization and single execution round. 3.**Dynamic structure with both spatial and temporal optimization without semantic**, as **Agent-Prune**(Zhang et al., 2024) and **AgentDropout**(Wang et al., 2025b), which are dynamic optimization method with multi-execution rounds based on policy-gradient using similar spatial-temporal graph without considering semantics. Particularly, to ensure the configuration consistency, we select AgentPrune-R for AgentPrune, which applies AgentPrune method on a random graph. These baselines include the main methods of optimizing topologies in the field of MAS.

### 4.1.3. IMPLEMENTATION DETAILS

For reasoning tasks, we set the number of dialogue rounds to $K = 2$, while for code-generation tasks on HumanEval we use $K = 4$. To ensure each agent in experiments have a different role, we set a prompt set for each domain of dataset, which would constrain the output of LLMs. For all datasets, we use five gpt-4o-based agents and keep the same role assignments and prompt sets across all methods to ensure a fair comparison. For the code generation dataset (HumanEval), we report pass@1, i.e., whether the first generated program passes all test cases. For all other reasoning datasets, we report standard accuracy. We use 40 samples to train the Critic (i.e., $N_c = 40$).

## 4.2. Performance & Cost Comparison (Q1)

### 4.2.1. PERFORMANCE

**Overall performance.** Table 1 reports the average accuracy of 91.36% for AgentTailor, surpassing AgentDropout by 0.67%, AgentPrune-R by 1.95%, GPTSwarm by 3.93%, and AutoGen by 9.71%. This indicates that semantic-aware edge judgment plus virtual execution consistently outperforms purely structural or temporal topology optimization. **By task type.** On general reasoning datasets (MMLU 86.14%), AgentTailor improves over AgentDropout (84.24%) by 1.90%, showing that semantic-guided training helps filter low-utility edges and boosts accuracy. As to

*Table 1.* Performance comparison between AgentTailor and three types of baselines. All multi-agent methods utilize five gpt-4o-based agents.

| Dataset | AutoGen | GPTSwarm | AgentPrune-R | AgentDropout | AgentTailor |
|---|---|---|---|---|---|
| Semantic aware | ✗ | ✗ | ✗ | ✗ | ✓ |
| HumanEval | 71.68 | 88.49 (+16.81) | 90.20 (+18.52) | 90.30 (+18.62) | 90.31 (+18.63) |
| GSM8K | 87.18 | 89.74 (+2.56) | 95.72 (+8.54) | 95.63 (+8.45) | 93.81 (+6.63) |
| MMLU | 82.35 | 83.98 (+1.63) | 83.94 (+1.59) | 84.24 (+1.89) | 86.14 (+3.79) |
| AQuA | 70.34 | 78.16 (+7.82) | 78.60 (+8.26) | 84.09 (+13.75) | 87.34 (+17.00) |
| MultiArith | 93.15 | 97.84 (+4.69) | 96.30 (+3.15) | 98.13 (+4.98) | 98.76 (+5.61) |
| SVAMP | 85.20 | 86.42 (+1.22) | 91.68 (+6.48) | 91.74 (+6.54) | 91.80 (+6.60) |
| Average | 81.65 | 87.43 (+5.78) | 89.41 (+7.76) | 90.69 (+9.04) | 91.36 (+9.71) |

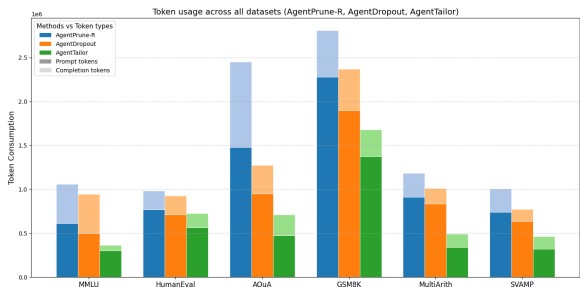

*Figure 2.* Token cost comparison between AgentTailor and other baseline methods.

mathematical reasoning datasets, AgentTailor performs well on most of them (SVAMP 91.80%, MultiArith 98.76% and AQuA 87.34%), outperforming AgentDropout by 0.06%, 0.63% and 3.25% respectively. Although AgentTailor shows a 1.91% decrease on GSM8K compared to AgentPrune-R, it still achieves better average performance across mathematical reasoning datasets.

**Variance and stability.** Table 2 complements the headline means in Table 1 with per-method standard deviations. Interpreting stability through the implied variance $\sigma^2$ (squared percentage points), GPTSwarm often attains the lowest spread on MMLU, HumanEval, and GSM8K, which aligns with its fixed single-round spatial graph and reduced structural randomness, even when its mean accuracy trails AgentTailor. AgentTailor reaches the best mean on MMLU, AQuA, and MultiArith, indicating repeatable outcomes under our evaluation. Conversely, temporal methods pay a stability cost on harder tasks—notably AgentPrune-R's std of 1.67 on HumanEval—while AgentTailor's std remains moderate relative to these peak as Table 1 shows.

*Table 2.* Mean accuracy (%) and standard deviation (percentage points) over independent RNG seeds for four multi-agent methods (AutoGen omitted; static baseline without the same multi-seed export). "avg" = mean accuracy, "std" = std. dev. of accuracy.

| Dataset | AgentTailor avg | AgentTailor std | AgentPrune-R avg | AgentPrune-R std | AgentDropout avg | AgentDropout std | GPTSwarm avg | GPTSwarm std |
|---|---|---|---|---|---|---|---|---|
| HumanEval | 90.31 | 1.21 | 90.21 | 1.670 | 91.3 | 1.340 | 88.49 | 0.690 |
| GSM8K | 93.80 | 0.812 | 95.72 | 0.790 | 95.63 | 0.560 | 89.74 | 0.550 |
| MMLU | 86.14 | 1.069 | 83.94 | 1.133 | 84.24 | 0.587 | 83.98 | 0.375 |
| AQuA | 87.34 | 1.077 | 78.60 | 0.546 | 84.09 | 0.461 | 78.16 | 0.546 |
| MultiArith | 98.76 | 0.00 | 96.30 | 0.210 | 98.13 | 0.440 | 97.84 | 0.310 |
| SVAMP | 91.79 | 0.46 | 91.61 | 1.270 | 91.74 | 1.070 | 86.42 | 0.710 |

#### 4.2.2. TOKEN COST ANALYSIS

Figure 2 demonstrates significant token efficiency improvements achieved by AgentTailor and shows that strong performance of temporal-optimized baselines comes at substantially higher token cost. AgentPrune-R and AgentDropout perform well across datasets by introducing temporal optimization and execution rounds, getting accuracy increment

with total token cost becomes 1.2–3.4× that of graph structure with single executing round (as GPTSwarm). Compared to AgentDropout, AgentTailor achieves reductions of 21.2%–61.6% in total tokens. Compared to AgentPrune-R, the reductions are 26.1%–70.9%.

When combined with GPTSwarm, AgentTailor reduces total token consumption by 27.8%–58.2% across different datasets, with completion token reductions of 49.8%–70.5% without accuracy decrease. These improvements highlight the effectiveness of AgentTailor's semantic-aware optimization policy in reducing token consumption through adaptive topology optimization and dynamic probability models.

### 4.3. Robustness Verification(Q2)

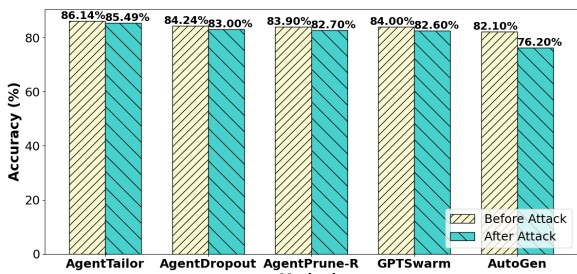

*Figure 3.* Different Methods' average accuracy before and after attack of 1 misleading agent

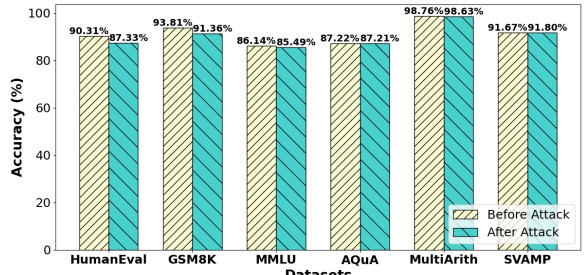

*Figure 4.* AgentTailor's accuracy before and after adversarial attacks across six datasets

We evaluate robustness by randomly replacing one agent with an adversarial agent that generates misleading output.

**Robustness Over Baselines.** As shown in Figure 3, AgentTailor demonstrates the least adversarial degradation among four different structures, with an average decrease of only 0.75%, much lower than AgentDropout(-1.24%), AgentPrune-R (-1.20%), GptSwarm(-1.40%) and AutoGen (-5.90%). Edge pruning and probabilistic scheduling allow its accuracy to remain almost unchanged before and after the attack (86.14% vs. 85.49%), demonstrating that dynamic pruning effectively mitigates the spread of malicious information.

**Cross-Dataset Robustness.** Figure 4 presents the results for the six benchmarks: minimal degradation on human code and general tasks (MMLU: -0.75%, HumanEval: -2.98%), with almost no impact on arithmetic-based benchmarks (AQuA, SVAMP and MultiArith nearly no decline), while GSM8K shows a -2.45% decrease. The experiments indicate that the model's performance remains stable and reproducible under attack. The results of Robustness experiments show that AgentTailor can actively reconstruct communication subgraphs during inference and block the spread of malicious information, compared to static topologies.

### 4.4. Experimental Analysis(Q3)

#### 4.4.1. ANALYSIS ON EPN

The Edge Prediction Network (EPN) approximates Edge-Judge supervision $\Delta_{ij}$ with a compact regressor so that virtual execution can rank edges without LLM calls. Evaluating this module by a single locking loss threshold alone would obscure how the per-edge summands of $\mathcal{L}_{\text{EPN}}$ (Eq. (10)) distribute across edges and whether predictions track the target signal or behave like noise. We therefore report the *full* distribution of squared residuals $(\hat{\Delta}_{ij} - \Delta_{ij})^2$ on the same pool of executed edges used for Critic training diagnostics, together with the Pearson correlation between $\hat{\Delta}_{ij}$ and $\Delta_{ij}$.

*Table 3.* EPN fidelity: correlation with Edge-Judge targets and distribution of per-edge squared residuals (summands of $\mathcal{L}_{\text{EPN}}$).

| Statistic | Value |
|---|---|
| Pearson $r$ ($\hat{\Delta}$ vs. $\Delta$) | 0.4769 |
| Mean $(\hat{\Delta} - \Delta)^2$ | 0.015 907 |
| Median $(\hat{\Delta} - \Delta)^2$ | 0.014 022 |
| P10 | 0.006 66 |
| P25 | 0.010 31 |
| P75 | 0.020 08 |
| P90 | 0.026 75 |

Table 3 lists the empirical fidelity statistics; we comment on their implications directly beneath the table. The mean and median squared residuals are close, suggesting similar-scale per-edge contributions to $\mathcal{L}_{\text{EPN}}$ rather than a few pathological outliers dominating the average. The P10–P90 spans

roughly 0.0067–0.0268, which suggests a moderate spread without extremely heavy upper tails at this resolution. Together with Pearson $r = 0.4769$, this pattern is more consistent with a moderate trend-following regressor than with random noise; yet the correlation is far from 1.0—so EPN should serve only as a guiding, coarse signal for ranking and virtual exploration. In practice, virtual execution aggregates predicted rewards over many sampled edges within each policy-gradient update, which can partially average out high-variance residuals but does not remove systematic bias; we therefore freeze EPN once $\mathcal{L}_{\text{EPN}}$ falls below $\tau_{\text{lock}}$ and continue grounding learning on Edge-Judge targets whenever real LLM executions are available.

#### 4.4.2. ABLATION STUDY

*Table 4.* Ablation study: total token consumption. "Before" denotes the full AgentTailor with virtual execution enabled; "After" denotes the variant without virtual execution.

| Dataset | Before | After | $\Delta$ (After–Before) |
|---|---|---|---|
| AQuA | 724299.8 | 920946.2 | +196646.4 |
| MultiArith | 490495.4 | 542917.6 | +52422.2 |
| SVAMP | 480706.0 | 546159.8 | +65453.8 |

*Table 5.* Ablation study: accuracy. "Before" denotes the full AgentTailor with virtual execution enabled; "After" denotes the variant without virtual execution.

| Dataset | Before | After | $\Delta$ (After–Before) |
|---|---|---|---|
| HumanEval | 90.31 | 87.33 | -2.98 |
| GSM8K | 93.81 | 92.28 | -1.53 |
| MMLU | 86.14 | 84.24 | -1.90 |
| AQuA | 87.34 | 87.21 | -0.13 |
| MultiArith | 98.76 | 98.63 | -0.13 |
| SVAMP | 91.80 | 91.33 | -0.47 |

**Ablation of Virtual Execution** Table 5 shows that removing virtual execution leads to accuracy degradation across most datasets, with the most significant drops in complex tasks: HumanEval (-2.98%), MMLU (-1.90%), and GSM8K (-1.53%). In contrast, simpler math tasks (AQuA, MultiArith, SVAMP) show minimal changes (<0.5%), suggesting that **virtual execution is crucial for tasks requiring extensive structure exploration.** Table 4 further reveals that ablating virtual execution increases token consumption by 52,422–196,646 tokens per dataset, demonstrating **virtual execution has dual benefit of improving accuracy while reducing computational cost.**

**Ablation of Relative-contribution normalization** The Edge Judge compares each edge to the final aggregated message and then converts raw cosine utilities into **relative contributions** $\Delta_{ij}$ to reduce *positional bias* (Eq. (2)). We ablate this normalization by removing the relative-contribution step and scoring edges with **raw cosine utility only**, keeping all other training and inference settings fixed. Table 6

reports three independent runs (RN-1–RN-3), their mean (RN-Avg), and the full AgentTailor model with normalization on MMLU and AQuA (cf. Eq. (2)).

*Table 6.* Ablation removing relative-contribution normalization ("RN": raw cosine edge utility only). RN-Avg averages RN-1–RN-3; **AgentTailor** is the full model with normalization. Accuracy in %; token counts sum prompt + completion where applicable.

| Dataset | Metric | RN-1 | RN-2 | RN-3 | RN-Avg | AgentTailor |
|---------|--------|------|------|------|--------|-------------|
| MMLU | Accuracy (%) | 84.38 | 84.46 | 85.62 | 84.82 | 86.14 |
| | Prompt tokens | 433,939 | 422,615 | 417,116 | 424,556 | 302,772 |
| | Completion tokens | 78,529 | 73,646 | 64,827 | 72,334 | 60,385 |
| AQuA | Accuracy (%) | 82.25 | 81.07 | 82.84 | 82.05 | 87.34 |
| | Prompt tokens | 695,649 | 708,185 | 861,656 | 755,163 | 471,104 |
| | Completion tokens | 273,600 | 278,893 | 314,423 | 288,972 | 242,202 |

Removing normalization yields a clear accuracy drop, especially on AQuA ($\approx$5.3% below the full model relative to RN-Avg) and a milder but consistent degradation on MMLU ($\approx$1.3%). Token usage also increases for the RN runs on both datasets (higher prompt and completion counts than AgentTailor), indicating that the bias-corrected signal not only improves mean quality but also supports sparser, more effective communication during training and inference. Overall, the experiment indicates that **relative-contribution normalization addresses positional bias rather than merely rescaling cosine scores**: the raw-utility alternative underperforms on both accuracy and efficiency.

### 4.4.3. SENSITIVITY ANALYSIS

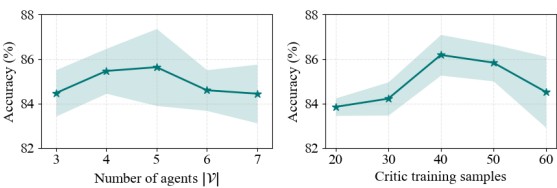

*Figure 5.* Sensitivity of AgentTailor to the number of agents (left) and the number of Critic training samples (right).

Figure 5 presents the sensitivity analysis of AgentTailor to two key hyperparameters: the number of agents $|\mathcal{V}|$ and the number of Critic training samples (early stopping round $N_c$).

**Sensitivity to the number of agents.** As shown in the left panel of Figure 5, accuracy increases from 84.5% in $|\mathcal{V}| = 3$ to a peak of 85.7% in $|\mathcal{V}| = 5$, then decreases to 84.7% and 84.5% in $|\mathcal{V}| = 6$ and $|\mathcal{V}| = 7$, respectively. This indicates that AgentTailor can exploit most collaborative benefits with a moderate team size of 5 agents, while larger teams introduce diminishing returns or even performance degradation, possibly due to increased communication overhead and redundant information flow.

**Sensitivity to Critic training samples.** The right panel shows that accuracy improves from 84.0% in $N_c = 20$ to

a peak of 86.2% in $N_c = 40$, then decreases to 85.8% and 84.5% for $N_c = 50$ and $N_c = 60$, respectively. The initial improvement suggests that more training samples provide a more reliable edge-utility estimation for the Critic, enabling better structure learning. The performance decrease at larger $N_c$ values indicates that overly aggressive fitting of the Critic may lead to overfitting or reduced generalization, as the EPN becomes too specialized to the training distribution.

**Selection of hyper-parameters.** Based on this analysis, we adopt $|\mathcal{V}| = 5$ agents and $N_c = 40$ Critic training samples as our default configuration, which achieves near-optimal performance while maintaining computational efficiency and robustness across different types of problems.

## 5. Conclusion

This work demonstrates that semantic-aware communication optimization can simultaneously improve both accuracy and efficiency in LLM-based multi-agent systems. Our experimental evaluation across six diverse datasets reveals that AgentTailor not only achieves superior performance (91.36% average accuracy) but also maintains robustness under adversarial conditions. Compared to existing methods, AgentTailor reduces token consumption by 21.2%–61.6% (vs. AgentDropout) while outperforming baselines including AgentPrune-R, GPTSwarm, AutoGen and AgentDropout. Ablation studies further confirm that virtual execution via the Edge Prediction Network is particularly crucial for complex tasks, providing both performance gains and token savings.

## 6. Limitations

Our approach has several limitations that suggest directions for future work. First, the Edge Prediction Network (EPN) exhibits only moderate predictive power. Due to this moderate correlation, the signal from EPN is primarily used to provide heuristic guidance and auxiliary adjustments to the graph, rather than fully dictating the graph generation process. Second, the method is primarily effective on problems that feature rich linguistic and textual content; its transfer to other domains may be unsatisfactory, because the aggregated results and the information passed along edges may not provide sufficiently discriminative evidence for the encoder. Third, introducing additional learnable graph structure may introduce variance into the system. Fourth, regarding robustness, we only analyze the setting where one agent is replaced by an adversarial agent; for richer attack models, effectiveness should be verified cautiously in realistic deployments . Addressing these issues, particularly improving edge prediction reliability and controlling the variance induced by structural learning, will be an important goal for future research.

## Acknowledgments

This work was supported by the National Natural Science Foundation of China under Grant No. 62302121.

## Impact Statement

This paper presents work whose goal is to advance the field of Machine Learning. There are many potential societal consequences of our work, none which we feel must be specifically highlighted here.

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

# A. Notation table

*Table 7.* Notation used in AgentTailor.

| Symbol | Description |
|---|---|
| $\mathcal{G} = (\mathcal{V}, \mathcal{E}_S, \mathcal{E}_T)$ | Actor graph with nodes and spatial/temporal edges |
| $\mathcal{V}$ | Set of agents (nodes) |
| $\mathcal{E}_S$ | Set of spatial edges (within-round communication) |
| $\mathcal{E}_T$ | Set of temporal edges (cross-round dependencies) |
| $e_{ij}$ | Directed edge from node $i$ to node $j$ |
| $\mathcal{E}_i$ | Incoming edges to node $i$ |
| $q$ | Task query / problem input |
| $K$ | Number of execution rounds |
| $d$ | Sentence embedding dimension (used by the encoder) |
| $K_v$ | Number of virtual execution steps per training/evaluation sample |
| $K_v^{(3)}$ | Number of virtual tailoring steps per query in Stage 3 |
| $N_1, N_2$ | Stage-1 / Stage-2 sample counts (training records) |
| $\mathbf{h}_{ij}$ | Embedding of the message on edge $e_{ij}$ |
| $\mathbf{h}_{\text{final}}$ | Embedding of the final aggregated message in a round |
| $U_{ij}$ | Utility of edge $e_{ij}$ (cosine similarity to final aggregated message) |
| $\Delta_{ij}$ | Relative contribution of edge $e_{ij}$ |
| $r_{ij}$ | Reward of edge $e_{ij}$ in real execution |
| $\hat{\Delta}_{ij}$ | Predicted relative contribution by EPN |
| $\hat{r}_{ij}$ | Reward in virtual execution |
| $\epsilon$ | Small constant for numerical stability in reward normalization |
| $\tau_{\text{lock}}$ | Locking threshold on Critic loss (Section 3.3) |
| $\tau_{\text{dag}}$ | Temperature for DAG-constrained edge sampling (Alg. 1) |
| $T$ | Temperature scaling for EPN output (Eq. (9)) |
| $\text{acc}_{\text{current}}$ | Current accuracy signal used for adaptive virtual learning rate |
| $\alpha_{\text{actor}}$ | Actor learning rate |
| $\alpha_{\text{virtual}}$ | Virtual execution learning rate (Eq. (14)) |
| $\alpha_{\text{min}}$ | Minimum virtual learning rate |
| $\lambda_2$ | Scaling factor for adaptive virtual learning rate |
| $\lambda_{\text{sparsity}}$ | Sparsity regularization strength |
| $l_{ij}$ | Logit of edge $e_{ij}$ |
| $p_{ij} = \sigma(l_{ij})$ | Activation probability of edge $e_{ij}$ |
| $\theta_S, \theta_T$ | Actor parameters for spatial / temporal edges |
| $\phi$ | Parameters of the Edge Prediction Network (EPN) |
| $\mathcal{E}_{\text{executed}}$ | Executed edges in real runs |
| $\mathcal{E}_{\text{sampled}}$ | Sampled edges in virtual execution |
| $m_{ij}$ | Binary mask indicating whether edge $e_{ij}$ is active |
| $p$ | Pruning ratio (fraction of edges removed) |
| $N_c$ | Number of Critic (EPN) training samples |

# B. Extra Experiments

### B.0.1. API CALLS, WALL-CLOCK TIME, AND MONETARY COST.

Beyond token counts, we report the number of paid LLM API invocations, end-to-end wall-clock time, and estimated USD spend under the same agent configuration and model pricing as in our runs. Table 8 shows that monetary cost tracks API-call volume almost linearly: methods that issue more completions (AgentPrune-R, AgentDropout with multiple execution rounds) dominate both metrics, whereas AgentTailor stays in the same regime as GPTSwarm on calls while achieving higher accuracy (Table 1).

This appendix complements the main paper with two intervention studies aimed at validating whether high-$\Delta_{ij}$ edges track practically important communication paths: *reverse pruning* (remove predicted high-contribution edges) and *frozen-graph leave-one-edge-out* (LOEO), which compares removing top- versus bottom-$\Delta$ edges within a single fixed topology.

*Table 8.* Wall-clock time (s), API calls, and estimated cost (USD) on six benchmarks. AgentPrune-R and AgentDropout use multiple execution rounds; GPTSwarm uses a single real execution round after spatial optimization.

| Dataset | Metric | AgentTailor | AgentPrune-R | AgentDropout | GPTSwarm |
|---|---|---|---|---|---|
| MMLU | Wall-clock (s) | 3098 | 4690 | 3603 | 3456 |
| | API calls | 887 | 2123 | 2121 | 1150 |
| | Cost (USD) | 3.96 | 9.65 | 6.61 | 4.47 |
| HumanEval | Wall-clock (s) | 5949 | 7827 | 5411 | 2276 |
| | API calls | 1047 | 2681 | 1213 | 899 |
| | Cost (USD) | 9.52 | 14.69 | 11.90 | 4.80 |
| AQuA | Wall-clock (s) | 4001 | 5362 | 3814 | 4072 |
| | API calls | 703 | 1848 | 1161 | 1106 |
| | Cost (USD) | 7.39 | 16.74 | 13.67 | 7.75 |
| GSM8K | Wall-clock (s) | 4028 | 4412 | 5369 | 3791 |
| | API calls | 837 | 2123 | 1413 | 1151 |
| | Cost (USD) | 7.51 | 19.88 | 14.40 | 9.39 |
| SVAMP | Wall-clock (s) | 2846 | 4341 | 2988 | 2306 |
| | API calls | 660 | 1763 | 1449 | 961 |
| | Cost (USD) | 4.04 | 9.20 | 8.47 | 4.52 |
| MultiArith | Wall-clock (s) | 3080 | 4492 | 5550 | 2402 |
| | API calls | 760 | 1760 | 1771 | 966 |
| | Cost (USD) | 4.63 | 7.61 | 12.40 | 4.29 |

## B.1. Reverse pruning

Standard pruning removes edges predicted to be low contribution. We instead perform **reverse pruning**: we drop edges with the *largest* predicted $\Delta_{ij}$, holding all other training and inference settings fixed.

Table 9 summarizes three independent reverse-pruning runs (RP-1–RP-3), their mean (RP-Avg), and the full AgentTailor model on MMLU and AQuA. Across both datasets, accuracy drops relative to the full model and prompt/completion token counts rise, indicating that high-$\Delta_{ij}$ edges indeed carry critical signal rather than noise—providing direct causal evidence aligned with our Edge-Judge construct.

*Table 9.* Reverse pruning: removing edges with highest predicted $\Delta_{ij}$ (three runs and average). **AgentTailor** = full model.

| Dataset | Metric | RP-1 | RP-2 | RP-3 | RP-Avg | AT |
|---|---|---|---|---|---|---|
| MMLU | Acc. (%) | 85.62 | 85.49 | 84.97 | 85.36 | 86.14 |
| | Prompt | 319,554 | 398,333 | 412,675 | 376,854 | 302,772 |
| | Compl. | 70,108 | 104,546 | 102,080 | 92,244 | 60,385 |
| AQuA | Acc. (%) | 82.25 | 84.62 | 82.74 | 83.20 | 87.34 |
| | Prompt | 616,784 | 579,183 | 574,204 | 590,057 | 471,104 |
| | Compl. | 254,283 | 259,325 | 273,935 | 262,514 | 242,202 |

## B.2. Frozen-graph leave-one-edge-out (LOEO)

Reverse pruning removes multiple high-$\Delta$ edges jointly. To reduce confounding from topology changes and isolate *per-edge* effects, we run a **frozen-graph LOEO** study. After training on $N_c=40$ samples, we sample *one* fixed communication graph. From that graph we identify the five edges with the largest $\Delta_{ij}$ (top-5) and the five with the smallest $\Delta_{ij}$ (bottom-5). For each selected edge we evaluate performance by removing **only that edge** while keeping all other edges identical. Evaluation uses a held-out test subset drawn from the same experimental pool as the main paper.

Table 10 reports three independent runs for each regime (Remove_top5_$\Delta$ vs. Remove_bottom5_$\Delta$). Accuracy is similar across the two regimes—consistent with single-edge removal being a mild perturbation when most of the graph remains active. However, both prompt and completion token counts are *consistently higher* when a top-$\Delta$ edge is removed than when a bottom-$\Delta$ edge is removed (matched within run index). Together with reverse pruning, this asymmetry suggests that high-$\Delta$ edges provide a **more token-efficient communication channel**: eliminating them forces the system to compensate through longer contexts or generations even when headline accuracy barely moves. The frozen-graph protocol

further ensures that comparisons are not driven solely by differing edge centralities across independently sampled graphs.

*Table 10.* Frozen-graph LOEO: one top- vs. bottom-$\Delta$ edge removed per trial (three runs). Acc. in %.

| Cond. | Metric | R1 | R2 | R3 |
|---|---|---|---|---|
| top5 $\Delta$ | Acc. (%) | 79.0 | 81.0 | 81.5 |
| | Prompt | 1919.9 | 1311.5 | 1703.0 |
| | Compl. | 311.2 | 313.2 | 455.0 |
| bot5 $\Delta$ | Acc. (%) | 79.5 | 81.0 | 83.0 |
| | Prompt | 1828.8 | 1026.4 | 1603.0 |
| | Compl. | 299.1 | 196.3 | 405.0 |

*Table 11.* Token and accuracy comparison of AgentPrune-R, AgentTailor, GPTSwarm, and GPTSwarm+AgentTailor on MMLU, HumanEval, AQuA.

| Dataset | Metric | AgentPrune-R | AgentDropout | AgentTailor | GPTSwarm | GPTSwarm+ AgentTailor |
|---|---|---|---|---|---|---|
| MMLU | Accuracy (%) | 83.94 | 84.24 | 86.14 | 83.98 | 84.67 |
| | Prompt tokens | 609,498 | 495,644 | 302,772 | 391,248 | 231,927 |
| | Completion tokens | 448,996 | 449,301 | 60,385 | 92,400 | 37,267 |
| HumanEval | Accuracy (%) | 90.20 | 90.30 | 90.31 | 88.49 | 88.82 |
| | Prompt tokens | 771,054 | 710,992 | 564,579 | 527,876 | 248,449 |
| | Completion tokens | 214,124 | 212,869 | 163,167 | 225,120 | 66,509 |
| AQuA | Accuracy (%) | 78.60 | 84.09 | 87.34 | 78.16 | 86.98 |
| | Prompt tokens | 1,479,527 | 952,166 | 471,104 | 424,332 | 376,605 |
| | Completion tokens | 974,106 | 324,043 | 242,202 | 319,353 | 160,378 |
| GSM8K | Accuracy (%) | 95.72 | 95.63 | 93.81 | 89.74 | 92.39 |
| | Prompt tokens | 2,281,863 | 1,900,032 | 1,372,904 | 959,317 | 558,618 |
| | Completion tokens | 527,890 | 469,873 | 307,592 | 268,926 | 117,745 |
| | Total tokens | 2,809,753 | 2,369,905 | 1,680,496 | 1,228,243 | 676,363 |
| MultiArith | Accuracy (%) | 96.30 | 98.13 | 98.76 | 97.84 | 97.51 |
| | Prompt tokens | 911,710 | 833,883 | 340,639 | 216,512 | 196,728 |
| | Completion tokens | 272,904 | 179,073 | 149,856 | 130,460 | 89,744 |
| | Total tokens | 1,184,614 | 1,012,956 | 490,495 | 346,972 | 286,472 |
| SVAMP | Accuracy (%) | 91.68 | 91.74 | 91.80 | 86.42 | 88.81 |
| | Prompt tokens | 740,593 | 635,114 | 318,451 | 224,826 | 195,965 |
| | Completion tokens | 265,211 | 139,906 | 147,401 | 140,027 | 87,228 |
| | Total tokens | 1,005,804 | 775,020 | 465,852 | 364,853 | 283,193 |

Table 11 demonstrate significant token efficiency improvements achieved by AgentTailor. Compared to AgentPrune-R, AgentTailor achieves reductions of 26.1%–70.9% in total tokens and 23.8%–86.6% in completion tokens across all six datasets, while maintaining or improving accuracy. When combined with GPTSwarm, AgentTailor further reduces token consumption by 17.4%–58.2% in total tokens and 31.2%–70.5% in completion tokens compared to GPTSwarm alone, with accuracy improvements on most datasets (e.g., HumanEval: 88.49% → 88.82%, AQuA: 78.16% → 86.98%). These improvements highlight the effectiveness of AgentTailor's semantic-aware optimization policy in reducing token consumption through adaptive topology optimization and dynamic probability models, while maintaining or even improving task performance across diverse domains.

# C. Reproducibility Details

## C.1. DAG Sampling

---

**Algorithm 1** DAG-Constrained Edge Sampling

---

**Require:** Node set $\mathcal{V}$, candidate edge set $\mathcal{E}$, logits $\{l_{ij}\}_{(i,j)\in\mathcal{E}}$, temperature $\tau_{\mathrm{dag}}$
**Ensure:** Active edge set $\mathcal{E}_{\mathrm{active}}$ forming a DAG
 1: Initialize $\mathcal{E}_{\mathrm{active}} \leftarrow \emptyset$
 2: Compute activation probabilities $p_{ij} = \sigma(l_{ij}/\tau_{\mathrm{dag}})$ for all $(i,j) \in \mathcal{E}$
 3: Sample a Bernoulli variable $z_{ij} \sim \mathrm{Bernoulli}(p_{ij})$ for each edge $(i,j) \in \mathcal{E}$
 4: Let $\mathcal{E}_{\mathrm{cand}} \leftarrow \{e_{ij} \in \mathcal{E} \mid z_{ij} = 1\}$ {stochastically activated edges}
 5: Optionally shuffle $\mathcal{E}_{\mathrm{cand}}$ to avoid ordering bias
 6: **for** each candidate edge $e_{ij} \in \mathcal{E}_{\mathrm{cand}}$ **do**
 7:     Tentatively set $\mathcal{E}' \leftarrow \mathcal{E}_{\mathrm{active}} \cup \{e_{ij}\}$
 8:     **if** FORMSCYCLE$(\mathcal{V}, \mathcal{E}') = $ **false** **then**
 9:         Accept the edge: $\mathcal{E}_{\mathrm{active}} \leftarrow \mathcal{E}'$
10:     **else**
11:         Reject the edge to maintain the DAG constraint
12:     **end if**
13: **end for**
14: **return** $\mathcal{E}_{\mathrm{active}}$

---

## C.2. Training Pipeline

---

**Algorithm 2** Graph Execution of AgentTailor

---

**Require:** Actor graph $\mathcal{G} = (\mathcal{V}, \mathcal{E}_S, \mathcal{E}_T)$ with logits $\theta_S, \theta_T$, input query $q$, number of rounds $K$
**Ensure:** Final answer $\hat{y}$
 1: Initialize node states and context for all agents
 2: **for** $t = 1$ to $K$ **do**
 3:     Sample active edges using $p_{ij} = \sigma(l_{ij})$ under DAG constraint
 4:     **for** each agent $v \in \mathcal{V}$ in topological order **do**
 5:         Collect incoming messages from active predecessors
 6:         Call LLM with agent prompt, history and messages, update node state
 7:     **end for**
 8: **end for**
 9: Aggregate outputs at the decision node and return $\hat{y}$

---

**Algorithm 3** Real Execution with Semantic Edge Judgment

---

**Require:** Actor, Critic (EPN), dataset record $r$, number of rounds $K$
**Ensure:** Answer $\hat{y}$, edge contributions $\{\Delta_{ij}\}$
 1: Build task input $q$ from $r$
 2: Run Graph Execution (Alg. 2) to obtain $\hat{y}$ and all executed edges $\mathcal{E}_{\mathrm{exec}}$
 3: Evaluate $\hat{y}$ with dataset metric to get pass ratio and correctness
 4: **for** each executed edge $e_{ij} \in \mathcal{E}_{\mathrm{exec}}$ **do**
 5:     Encode edge message and final output message, compute utility $U_{ij}$ (Eq. (1))
 6: **end for**
 7: For each $e_{ij}$, compute relative contribution $\Delta_{ij}$ (Eq. (2))
 8: Normalize $\Delta_{ij}$ to rewards $r_{ij}$ (Eq. (3))
 9: Update Critic (EPN) using $\mathcal{L}_{\mathrm{EPN}}$ (Eq. (10))
10: Update Actor logits with $\mathcal{L}_{\mathrm{real}}$ (Eq. (4))

---

---

**Algorithm 4** Virtual Execution with Edge Prediction Network

---

**Require:** Actor with logits $\theta_S, \theta_T$, locked Critic (EPN) with parameters $\phi$, record $r$, number of rounds $K$, virtual steps $K_v$
**Ensure:** Updated Actor logits $\theta_S, \theta_T$
  1: Build task input $q$ from $r$
  2: **for** $k = 1$ to $K_v$ **do**
  3:     Sample active edges $\mathcal{E}_{\text{sampled}}$ using current $p_{ij} = \sigma(l_{ij})$
  4:     **for** each $e_{ij} \in \mathcal{E}_{\text{sampled}}$ **do**
  5:        Construct EPN input $\mathbf{x}_{ij}$ from source/target node descriptions, histories and $q$
  6:        Predict $\hat{\Delta}_{ij} = \text{EPN}_\phi(\mathbf{x}_{ij})$ (Eq. (11))
  7:     **end for**
  8:     Normalize $\hat{\Delta}_{ij}$ to virtual rewards $\hat{r}_{ij}$ (Eq. (12))
  9:     Set virtual learning rate $\alpha_{\text{virtual}} = \max(\lambda_2 \cdot \text{acc}_{\text{current}}, \alpha_{\min})$ (Eq. (14))
10:     Update Actor logits $\theta_S, \theta_T$ by minimizing $\mathcal{L}_{\text{virtual}}$ (Eq. (13))
11: **end for**

---

**Algorithm 5** Overall Training and Evaluation Logic of AgentTailor

---

**Require:** Dataset $\mathcal{D}$, Actor logits $\theta_S, \theta_T$, Critic (EPN) parameters $\phi$, rounds $K$, Stage-1/2 sample counts $N_1, N_2$, virtual steps $K_v$, Stage-3 virtual steps $K_v^{(3)}$, prune ratio $p$
**Ensure:** Trained Actor/Critic, pruned graph, final accuracy and token statistics
  1: Initialize Actor, Critic, and TrainingState
  2: **Stage 1 (Real-only): semantic supervision collection**
  3: **for** $n = 1$ to $N_1$ **do**
  4:     Sample record $r \sim \mathcal{D}$, build task input $q$
  5:     Run graph execution with LLM for $K$ rounds, obtain answer $\hat{y}$ and executed edges $\mathcal{E}_{\text{exec}}$
  6:     Edge Judge computes $\Delta_{ij}$ for each $e_{ij} \in \mathcal{E}_{\text{exec}}$
  7:     Update Critic $\phi$ to fit $\Delta_{ij}$; update Actor logits $\theta_S, \theta_T$ using real rewards
  8: **end for**
  9: **Stage 2 (Hybrid): virtual steps + one real step per sample**
10: **for** $n = 1$ to $N_2$ **do**
11:     Sample record $r \sim \mathcal{D}$, build task input $q$
12:     **for** $k = 1$ to $K_v$ **do**
13:        Run virtual execution: sample edges, predict $\hat{\Delta}_{ij}$ with EPN, update Actor logits without LLM calls
14:     **end for**
15:     Run one real execution as in Stage 1 to update both Actor and Critic
16: **end for**
17: Save Stage-2 snapshot of logits (and masks) for reproducible evaluation
18: **Stage 3 (Snapshot-based evaluation): per-query tailoring + pruning**
19: **for** each remaining evaluation record $r \in \mathcal{D}$ **do**
20:     Reset Actor to the Stage-2 snapshot state
21:     **for** $k = 1$ to $K_v^{(3)}$ **do**
22:        Run virtual execution to tailor edge probabilities for this query
23:     **end for**
24:     Prune bottom $p\%$ edges by probability; run real execution on the pruned graph and log metrics
25: **end for**
26: **return** final metrics, trained parameters, and pruned graphs

---

# D. Edge Judgment and Reward Computation

In the main text, we define the utility of an edge $e_{ij}$ by measuring how much its message is adopted in the final decision of the current round. Concretely, for each executed edge $e_{ij}$ we extract the textual message passed from agent $i$ to agent $j$ and

encode it with a shared sentence encoder into a fixed-dimensional vector

$$\mathbf{h}_{ij} = \text{Enc}(\text{message}_{ij}).$$

At the end of each round, we also extract the final aggregated message of that round, which corresponds to the input to the decision node (or the concatenation of all agents' outputs that are used to produce the final answer), and encode it as

$$\mathbf{h}_{\text{final}} = \text{Enc}(\text{final\_message}).$$

The semantic utility of edge $e_{ij}$ is then defined as the cosine similarity between these two embeddings:

$$U_{ij} = \cos(\mathbf{h}_{ij}, \mathbf{h}_{\text{final}}) = \frac{\mathbf{h}_{ij} \cdot \mathbf{h}_{\text{final}}}{\|\mathbf{h}_{ij}\| \, \|\mathbf{h}_{\text{final}}\|}. \tag{16}$$

Intuitively, $U_{ij} \approx 1$ means that the content on edge $e_{ij}$ is highly aligned with what is finally used to answer the query, whereas values near 0 or negative indicate that the message is irrelevant or even contradictory to the final decision.

**Relative contribution and position bias.** A direct use of $U_{ij}$ would be biased towards edges closer to the final node, because messages propagated later in the computation chain naturally have higher similarity to the final output. To remove this positional bias, we compare each outgoing edge with the average utility of the incoming edges to its source node. Let $\mathcal{E}_i = \{e_{ki}\}$ denote all edges pointing into node $i$. We first compute the average incoming utility

$$\bar{U}_i = \frac{1}{|\mathcal{E}_i|} \sum_{e_{ki} \in \mathcal{E}_i} U_{ki}, \tag{17}$$

and then define the relative contribution of edge $e_{ij}$ as

$$\Delta_{ij} = U_{ij} - \bar{U}_i. \tag{18}$$

This quantity measures how much additional semantic value the outgoing edge $e_{ij}$ contributes beyond what node $i$ has already received from its predecessors. A positive $\Delta_{ij}$ indicates that the edge carries novel, useful information, while a negative $\Delta_{ij}$ suggests redundancy or harmful content. For nodes without any incoming edges (e.g., source nodes), we set $\bar{U}_i = 0$ so that $\Delta_{ij} = U_{ij}$.

**Normalization into rewards.** To use relative contributions as reinforcement learning signals for the Actor, we normalize $\Delta_{ij}$ at the round level. For each round $t$, let $\mathcal{E}_t$ denote the set of edges that were actually executed. We define the reward for each executed edge as

$$r_{ij} = \frac{\Delta_{ij}}{\left| \sum_{e_{kl} \in \mathcal{E}_t} \Delta_{kl} \right| + \epsilon}, \tag{19}$$

where $\epsilon > 0$ is a small constant to avoid division by zero. This normalization keeps the sign of $\Delta_{ij}$ unchanged while scaling rewards to be comparable across different rounds and tasks. In particular, it ensures that (i) the magnitude of rewards is bounded by the total contribution in the round, and (ii) both strongly positive and strongly negative edges provide clear learning signals.

**Policy gradient and sparsity regularization.** The Actor parameterizes edge activation probabilities $p_{ij} = \sigma(l_{ij})$ via logits $l_{ij}$. Given the rewards above, we use a REINFORCE-style loss:

$$\mathcal{L}_{\text{real}} = -\frac{1}{|\mathcal{E}_{\text{executed}}|} \sum_{e_{ij} \in \mathcal{E}_{\text{executed}}} \log(p_{ij}) \cdot r_{ij}$$
$$+ \lambda_{\text{sparsity}} \cdot \left( \frac{1}{|\theta_S|} \sum_{l_{ij} \in \theta_S} |l_{ij}| + \frac{1}{|\theta_T|} \sum_{l_{ij} \in \theta_T} |l_{ij}| \right). \tag{20}$$

The first term encourages the Actor to increase the probability of high-contribution edges ($r_{ij} > 0$) and suppress low- or negative-contribution edges. The second term is an $\ell_1$ sparsity penalty on spatial and temporal logits, which explicitly pushes the communication graph towards a sparse topology.

**Practical considerations.** In practice, we only compute utilities for edges that are actually executed in a round, which keeps the cost of edge judgment linear in the number of active edges. We use a shared encoder across all messages and rounds to ensure that utilities are comparable across tasks and datasets. These design choices make semantic edge judgment both computationally tractable and stable enough to provide reliable supervision for the Critic and the Actor.

# E. EPN Architecture and Locking Mechanism

The Edge Prediction Network (EPN) is a lightweight multilayer perceptron that takes edge-level context as input and predicts the intrinsic value (or relative contribution) $\hat{\Delta}_{ij}$ for each candidate edge $e_{ij}$. In the implementation, the EPN is parameterized by a list of layer dimensions

$$\texttt{dims} = [d_0, d_1, \ldots, d_{L-1}],$$

where $d_0$ is the input dimension and $d_{L-1}$ is the output dimension. The input dimension $d_0 = 5 \times 384$ corresponds to the concatenation of five 384-dimensional sentence embeddings (source node description, source node history, task query, target node description, and target node history), while the output dimension $d_{L-1} = 1$ produces a scalar prediction. The intermediate dimensions $[d_1, \ldots, d_{L-2}]$ define a sequence of hidden layers that progressively transform the input representation.

**MLP backbone.** Given the dimension list $\texttt{dims}$, we define the hidden dimensions as $[d_0, \ldots, d_{L-2}]$ and use all but the last hidden size to construct the MLP backbone. For each consecutive pair $(d_\ell, d_{\ell+1})$ in the hidden dimensions, we add a fully connected layer with ReLU activation (and optional dropout):

$$\mathbf{h} = \text{MLP}(\mathbf{x}_{ij}), \tag{21}$$

where the MLP is a sequence of

$$\text{Linear}(d_\ell, d_{\ell+1}) \to \text{ReLU} \to (\text{optional Dropout}).$$

The dropout ratio is controlled by a hyperparameter $\texttt{dropout}$. The final hidden dimension $d_{L-2}$ is then used as the input size of the residual block and the output layer.

**Residual refinement block.** To improve expressiveness without significantly increasing depth, we add a lightweight residual block on top of the last hidden representation. Let $d_{\text{hid}} = d_{L-2}$ denote the last hidden size. The residual block is implemented as

$$\mathbf{r} = \text{Linear}_2(\text{GELU}(\text{Linear}_1(\mathbf{h}))), \tag{22}$$

where both $\text{Linear}_1$ and $\text{Linear}_2$ map $\mathbb{R}^{d_{\text{hid}}} \to \mathbb{R}^{d_{\text{hid}}}$. We optionally apply dropout to the residual branch with the same ratio $\texttt{dropout}$, and then add it back to the backbone output:

$$\mathbf{h}' = \mathbf{h} + \text{Dropout}(\mathbf{r}). \tag{23}$$

**Output layer and temperature scaling.** The final prediction is obtained by a linear projection followed by a temperature-scaled hyperbolic tangent:

$$\mathbf{o} = \mathbf{W}_o \mathbf{h}' + b_o, \qquad \hat{\Delta}_{ij} = \tanh(\mathbf{o} \cdot T), \tag{24}$$

where $\mathbf{W}_o \in \mathbb{R}^{1 \times d_{\text{hid}}}$, $b_o \in \mathbb{R}$, and $T > 0$ is a temperature hyperparameter. In code, the temperature is implemented as a scalar multiplier on the output ('output = torch.tanh(output * temperature)'). The $\tanh$ activation bounds predictions in $(-1, 1)$ and the temperature controls the contrast between high- and low-value edges.

**Input construction.** The EPN is used together with a sentence encoder (a pre-trained SentenceTransformer such as $\texttt{all-MiniLM-L6-v2}$). Given three input texts (source node information, target node information, and question), we encode each into a 384-dimensional embedding and concatenate them to obtain

$$\mathbf{x}_{ij} \in \mathbb{R}^{5 \times 384}.$$

In some ablations we also consider a five-part input (node descriptions and histories plus the query), in which case the input dimension becomes $5 \times 384$ and $\texttt{dims}$ is adjusted accordingly. Importantly, the sentence encoder is frozen during Critic training, and only the EPN parameters are updated.

**Training objective.** The EPN is trained to regress edge-level supervision (e.g., relative contributions or intrinsic edge values) using mean squared error when targeting continuous labels, or binary cross-entropy when targeting binary labels. Let $\mathcal{E}_{\text{executed}}$ denote the set of edges with available supervision. For continuous targets $\Delta_{ij}$ we use

$$\mathcal{L}_{\text{EPN}} = \frac{1}{|\mathcal{E}_{\text{executed}}|} \sum_{e_{ij} \in \mathcal{E}_{\text{executed}}} \left( \hat{\Delta}_{ij} - \Delta_{ij} \right)^2, \tag{25}$$

while for binary targets we use standard binary cross-entropy between predictions and labels.

# F. GSM8K Case Study: Commission Semantics Under Edge Pruning

This appendix documents a **pruned-vs.-unpruned flip** on a single GSM8K test item, using traces produced by our visualization script (`visualize_gsm8k_case_study.py`): same trained logits and decoding settings, **bottom-**$25\%$ **probability edges masked** in the pruned run and **full masks** in the unpruned run. The example is chosen because the **final monetary answer changes** between the two regimes while the **word problem text is identical**, which isolates the effect of *communication topology* on multi-agent aggregation.

## F.1. Problem statement

*A marketing company pays its employees on a commission-based salary system. If you sell goods worth $1000, you earn a 30% commission. Sales over $1000 get you an additional 10% commission. Calculate the amount of money Antonella earned if she sold goods worth $2500.*

**GSM8K reference solution.** On the first $1000 of sales the commission is $0.30 \times 1000 = 300$. On the excess $(2500-1000)=1500$, the *additional* commission is $0.10 \times 1500 = 150$. Total earnings: $300 + 150 = \mathbf{450}$ dollars.

## F.2. Observed graph structure and outputs

### Notation.

Concretely, its prompt is built from the **concatenation** of the round's eligible agent outputs—the same object encoded as $\mathbf{h}_{\text{final}}$ when defining edge utilities in Eq. (1)—and a **single** LLM pass then writes the final numeric line *without* an explicit majority-vote layer over competing numbers. In the pruned failure trace, the decision rationale therefore leans on the $900 side partly because that agent packages the misreading as **runnable code**, which the arbiter treats as higher-confidence evidence even when another visible branch states the GSM8K-consistent piecewise computation.

### F.2.1. PRUNED GRAPH

- **Round 0 (spatial). Math Solver** broadcasts to **Mathematical Analyst**, **Programming Expert**, **Inspector**, and **Math Solver** a single narrative: the first $1000 earns 30%, and the *remaining* $1500 is taxed at an *effective* $30\% + 10\% = 40\%$, yielding $300 + 600 = 900$.

- **Round 1 (spatial / temporal).** Several branches still derive the **piecewise** rule $300 + 0.10 \times 1500 = 450$ (notably after Answer Hints mention competing numbers). A Mathematical Analyst style derivation states $E = (1000 \cdot C_1) + ((S - 1000) \cdot C_2)$ with $C_1 = 0.30$, $C_2 = 0.10$, which matches the gold.

### F.2.2. UNPRUNED GRAPH

We further compare the reasoning trajectories produced by the pruned and unpruned communication graphs on the same GSM8K commission problem.

- **Round 0.** Several agents initially converge toward the incorrect **40% commission-rate interpretation**, as reflected in the intermediate outputs of Math Solver.

- **Round 1.** Under the unpruned graph, corrective signals propagate across multiple communication edges (e.g., Mathematical Analyst→Programming Expert). Multiple agents explicitly reconstruct the correct piecewise commission structure: 30% for the first sales bracket and an additional **10% only** for the excess portion, yielding the decomposition $300 + 150$.

- **Final decision.** The final aggregation step identifies consensus among agents on the value of **$**450. Although both interpretations of "additional" appear during intermediate reasoning, the system ultimately selects the incremental-10% interpretation consistent with the problem statement.

Table 12 summarizes the final outputs under the two communication settings. The comparison shows that communication pruning removes critical corrective interactions, causing the system to converge to the erroneous answer of 900, whereas the full communication graph preserves sufficient cross-agent verification to recover the correct solution.

*Table 12.* Effect of communication pruning on solution correctness for the GSM8K commission problem.

| Communication setting | Final answer (USD) | Correctness |
|---|---|---|
| Pruned graph ($p = 0.25$) | 900 | Incorrect |
| Full graph (unpruned) | 450 | Correct |

### F.3. Analysis and Discussion

**Linguistic ambiguity in the task formulation.** The phrase "*an additional* 10% commission" for sales exceeding $1000 admits two internally consistent formalizations: (i) the **gold interpretation**, in which the extra 10 percentage points apply only to the excess sales amount ($0.10 \times 1500$); and (ii) a **spurious interpretation**, in which the marginal commission rate is treated as 40% over the entire excess portion ($0.40 \times 1500$). Both interpretations can be supported by coherent multi-step derivations. Since GSM8K provides open-ended supervision rather than multiple-choice constraints, disambiguation must emerge entirely from the model's internal reasoning process.

**Effect of communication pruning.** Communication pruning removes low-probability edges prior to each round of DAG sampling, thereby reducing the number of reasoning trajectories available for cross-agent verification. In this example, the pruned graph still contains local traces of the correct piecewise decomposition. However, the early Math Solver outputs in round 0 predominantly favor the 40% interpretation, and this reasoning pattern is subsequently reinforced through code-oriented derivations that support the $900 solution. Because fewer communication paths survive pruning, corrective reasoning reaches the final decision node with reduced redundancy and weaker consensus signals.

By contrast, the unpruned graph preserves a larger set of spatial and temporal communication edges. As a result, the correct decomposition recurs across multiple independent reasoning branches in round 1, allowing consistent corrective evidence to accumulate and dominate the final aggregation process.

**Implications.** This case study yields two observations.

1. **Communication topology directly affects reasoning outcomes.** Under identical prompts and model parameters, modifying only the communication mask can lead to different final predictions by changing which intermediate arguments are propagated to the decision node and how frequently they are reinforced.
2. **GSM8K evaluates aggregation robustness in addition to arithmetic competence.** The failure mode observed here does not arise from arithmetic error, but from selecting an incorrect semantic interpretation despite the presence of correct intermediate derivations elsewhere in the graph.

## G. Datasets, Prompts, and Agent Roles

### G.1. Prompts for Complex Reasoning Tasks

The MMLU dataset uses five distinct agent roles, each with specific responsibilities for analyzing and solving multi-choice questions. The prompts for each role are shown in Figure 6.

### G.2. Prompts for Mathematical Reasoning Tasks

The four mathematical reasoning datasets (GSM8K, MultiArith, SVAMP, and AQuA) share a common set of four agent roles, each with specific responsibilities for solving mathematical problems. While the core role descriptions are consistent across these datasets, the output format differs: GSM8K, MultiArith, and SVAMP require numerical answers, while AQuA requires selecting from multiple-choice options (A, B, C, D, or E). The prompts for each role are shown in Figure 7:

### G.3. Prompts for Code Generation Tasks (HumanEval)

The HumanEval dataset uses a software engineering workflow with five distinct agent roles, each responsible for a specific stage of the code generation process: from design to implementation, testing, and bug fixing. The prompts for each role are shown in Figure 8.

### Prompt for Critic

You are an excellent critic. Please point out potential issues in Other agent's analysis point by point.

### Prompt for Historian

You research and analyze cultural, economic, political, and social events in the past, collect data from primary sources and use it to develop theories about what happened during various periods of history.

### Prompt for Knowlegable Expert

You are a knowlegable expert in question answering. Please give several key entities that need to be searched in wikipedia to solve the problem. Key entities that need to be searched are included between two '@' when output, for example: @catfish effect@, @broken window effect@, @Shakespeare@. If there is no entity in the question that needs to be searched in Wikipedia, you don't have to provide it.

### Prompt for Mathematician

You are a mathematician who is good at math games, arithmetic calculation , and long-term planning.

### Prompt for Psychologist

You are a psychologist. You are good at psychology, sociology, And philosophy. You give people scientific suggestions that will make them feel better.

*Figure 6.* Prompt for Complex Reasoning Tasks

### Prompt for Inspector

You are an Inspector. You will be given a math problem, analysis and code from other agents. Check whether the logic/calculation of the problem solving and analysis process is correct(if present). Check whether the code corresponds to the solution analysis(if present). Give your own solving process step by step based on hints. The last line of your output contains only the final result without any units, for example: The answer is 145 .You will be given some examples you may refer to.

### Prompt for Math Solver

You are a math expert. You will be given a math problem and hints From other agents. Give your own solving process step by step based on hints. The last line of your output contains only the final result without any units, for example: The answer is 145 .You will be given some examples you may refer to.

### Prompt for Mathematical Analyst

You are a mathematical analyst. You will be given a math problem, analysis and code from other agents. You need to first analyze the problem- solving process step by step, where the variables are represented by letters. Then you substitute the values into the analysis process to perform calculations and get the results. The last line of your output contains only the Final result without any units, for example: The answer is 145 .You will be given some examples you may refer to.

### Prompt for Project Manager

You are a programming expert. You will be given a math problem, analysis and code from other agents. Integrate step-by-step reasoning and Python code to solve math problems. Analyze the question and write functions to solve the problem. The function should not take any arguments and use the final result as the return value. The last line of code calls the function you wrote and assigns the return value to the \(answer) variable. Use a Python code block to write your response. For example: python def fun():x = 10 y = 20 return x + y answer = fun() Do not include anything Other than Python code blocks in your response. You will be given some examples you may refer to.

*Figure 7.* Prompts for Mathematical Reasoning Tasks

### Prompt for Algorithm Designer

You are an algorithm designer. You will be given a function signature and its docstring by the user. You need to specify the specific design of the algorithm, including the classes that may be defined and the functions used. You need to generate the detailed documentation, including explanations of the algorithm, usage instructions, and API references. When the implementation logic is complex, you can give the pseudocode logic of the main algorithm. I hope your reply will be more concise. Preferably within fifty words. Don't list too many points.

### Prompt for Bug Fixer

You are a bug fixer. You will be given a function signature and its docstring by the user. You need to provide modified and improved python code based on the current overall code design, algorithm framework, code implementation or test problems. Write your full implementation (restate the function signature). Use a Python code block to write your response. For example: 'python print ('Hello world!') Do not include anything other than Python code blocks in your response Do not change function names and input variable types in tasks.

### Prompt for Normal Programmer

You are a programmer. You will be given a function signature and its docstring by the user. You can refer to the agents' outputs. Write your full implementation (restate the function signature). Use a Python code block to write your response. For example: "python print('Hello world!') Do not include anything other than Python code blocks in your response.

### Prompt for Programming Expert

You are a programming expert. You will be given a function signature and its docstring by the user. You may be able to get the output results of other agents. They may have passed internal tests, but they may not be completely correct. Write your full implementation (restate the function signature). Use a Python code block to write your response. For example: "python print('Hello world!')"Do not include anything other than Python code blocks in your response. Do not change function names and input variable types in tasks.

### Prompt for Project Manager

You are a project manager. You will be given a function signature and its docstring by the user. You are responsible for overseeing the overall structure of the code, ensuring that the code is structured to complete the task Implement code concisely and correctly without pursuing over-engineering. You need to suggest optimal design patterns to ensure that the code follows best practices for maintainability and flexibility. You can specify the overall design of the code, including the classes that need to be defined (maybe none) and the functions used (maybe only One function). I hope your reply will be more concise. Preferably within fifty words. Don't list too many points.

*Figure 8.* The first part of Prompts for Code Generation Tasks

