# OpenReview forum: "AgentTailor: A Semantic-Aware LLM-Based Multi-Agent System with Actor-Critic Structure"
_ICML.cc/2026/Conference — ICML 2026 regular_

### Official Review · Reviewer_YZ7b · 2026-03-09

**Soundness:** 2
**Presentation:** 2
**Significance:** 2
**Originality:** 2
**Overall Recommendation:** 2
**Confidence:** 3

**Summary:**

This article addresses the problem of efficiency and scalability in multi-agent systems using LLMs, with a particular focus on reducing redundant communications between agents. The authors propose AgentTailor, which evaluates the contribution of communication edges in the agent graph. AgentTailor employs an actor–critic architecture combined with an Edge Prediction Network to learn and dynamically optimize the communication graph while limiting calls to the LLM. The method is evaluated on six benchmarks covering different categories of tasks: MMLU, GSM8K, AQuA, MultiArith, SVAMP, and HumanEval. The results show very strong performance while reducing total token consumption by approximately 21% to 61% compared to the best existing multi-agent systems.

**Compliance With Llm Reviewing Policy:**

Affirmed.

**Key Questions For Authors:**

See previous comments

**Limitations:**

See previous comments

**Strengths And Weaknesses:**

The article addresses the issue of efficiency in multi-agent systems using language models by seeking to reduce the communication cost between agents. To achieve this, the authors propose computing the contribution of a communication between agent i and agent j using the cosine similarity between the text produced by agent i and the text corresponding to the final solution to the problem. This value is then approximated, without invoking the LLM, using a neural network. The approach is evaluated on several benchmarks covering different types of tasks (reasoning, mathematics, and code generation). The experimental results show a notable reduction in token consumption while maintaining, or even improving, accuracy compared to several state-of-the-art multi-agent systems.

Despite these contributions, several methodological aspects raise questions. First, the training of the Critic relies on a very limited number of examples (only a few dozen problems generating a few hundred supervised edges), which seems small given the dimensionality of the model inputs and the number of parameters in the MLP. Moreover, part of the Critic’s input variables (notably the agents’ role descriptions: source role and target role) is fixed in the experiments. What additional information do they actually bring to the data? Finally, the contribution of an edge is based on the cosine similarity between the produced text and the final answer, but is this assumption really valid?

- soundness

The main issue lies in the central assumption of the method: the importance of an edge in the agent graph is estimated from the cosine similarity between the transmitted message and the final answer (or more precisely from a relative contribution derived from this similarity). However, this assumption is not experimentally validated in the paper. It would have been interesting for the authors to explicitly test the relevance of this metric. For instance, they could remove the edges considered the most important and verify whether this indeed leads to a degradation in performance. Conversely, removing edges deemed less important should, according to this hypothesis, have only a limited impact on the system’s performance.
Furthermore, among the inputs of the neural network used to estimate the importance of an edge without relying on the LLM, some variables are static and do not depend on the problem being solved but only on the architecture of the multi-agent system, such as the role of the source agent and that of the target agent. Since these roles remain fixed in the experiments, they do not introduce variation across training examples. However, the authors do not discuss why these pieces of information are included in the model inputs nor what influence they may have on the learning process. It is possible that the model mainly learns structural priorities between agent types rather than an estimation that truly depends on the problem.
Finally, the neural network used to approximate the importance of edges appears to be trained on a relatively small dataset. Yet the model inputs are high-dimensional and the MLP may contain a large number of parameters. In this context, it would be useful for the authors to provide more details about the actual amount of supervised data used to train this model, as well as the mechanisms used to prevent overfitting. The paper also does not discuss why such a model would be expected to generalize well from such a limited number of examples.

- Presentation

The paper is generally well written and provides numerous technical details about the proposed method (loss functions, activation functions, data representations, algorithm pseudocode, etc.), which makes it possible to understand the overall functioning of the approach. Nevertheless, some methodological aspects could be described more precisely to facilitate the full reproducibility of the experiments, particularly the exact amount of data used to train the Critic, certain hyperparameter choices, and the detailed functioning of the virtual execution step during inference. The authors also indicate that the code will be made publicly available if the paper is accepted, which should help clarify these ambiguities and facilitate the reproduction of the results.

- Significance

The objective pursued by the authors is both relevant and important. Multi-agent systems based on LLMs require numerous calls to these models, which results in significant computational and financial costs. In this context, developing methods that limit unnecessary communication between agents and reduce the use of LLM calls represents an important challenge for making such systems more efficient and easier to deploy.

- Originality

The problem addressed, namely reducing the use of LLMs in a multi-agent system, as well as the type of solution proposed, consisting of pruning the communication graph between agents, are approaches that have already been widely explored in recent years. The originality of the paper mainly lies in the metric used to measure the importance of an edge and in the use of an MLP to approximate this value without relying on the LLM. Nevertheless, despite the strong experimental results reported, this approach raises several methodological questions.

---

> ### Author Rebuttal · Authors · 2026-03-30
>
> We sincerely thank the reviewer for the critical and insightful comments. These questions help us further clarify the core assumptions and generalization properties of our method. We provide additional empirical evidence and detailed explanations to support the validity of our design.
>
> For the question of the validity of cosine similarity , we directly validate the cosine-similarity-based utility proxy through reverse pruning experiments. Removing edges with the highest predicted contribution leads to significant performance degradation, demonstrating that these edges are indeed crucial for final predictions. This provides empirical support that the proxy captures meaningful semantic alignment and reflects true edge contribution.
>
> **TABLE 1: Experiment with Reversed-Pruning**
> | Dataset | Metric            | RP-1   | RP-2   | RP-3   | Avg    | AgentAC |
> |---------|------------------|--------|--------|--------|--------|---------|
> | mmlu    | Accuracy         | 85.62  | 85.49  | 84.97  | 85.36  | 86.14   |
> |         | Prompt-token     | 319,554| 398,333| 412,675| 376,854| 302,772 |
> |         | Completion-token | 70,108 | 104,546| 102,080| 92,244 | 60,385  |
> | aqua    | Accuracy         | 82.25  | 84.62  | 82.74  | 83.20  | 87.34   |
> |         | Prompt-token     | 616,784| 579,183| 574,204| 590,057| 471,104 |
> |         | Completion-token | 254,283| 259,325| 273,935| 262,514| 242,202 |
>
> For the question on EPN training, although the Critic is trained from N_c = 40 problems, supervision is constructed at the edge level, yielding several hundred labeled edges. In addition, the Critic is a lightweight MLP designed to learn coarse signals rather than memorize patterns, which reduces overfitting risk. We also observe consistent downstream gains across tasks, indicating that the model captures transferable contribution patterns rather than overfitting to limited data.
>
> For the question about static feature inputs in the EPN, We thank the reviewer for this observation regarding the static feature inputs in the EPN. We would like to clarify that these features are not used to distinguish between different problem instances; rather, they serve as a structural prior to help the model capture general patterns of interaction between agent roles. Specifically, the instance-specific importance of edges is primarily determined by dynamic features (e.g., intermediate messages and interaction history), which vary across problem instances. Therefore, the model does not rely solely on static features for decision-making and is still able to effectively capture instance-level variations.

---

> > ### Author Rebuttal · Reviewer_YZ7b · 2026-04-05
> >
> > The rebuttal does not give satisfactory answers to all my remarks and the justifications of the choices and metrics should have been in the submitted paper.

---

### Official Review · Reviewer_6rg9 · 2026-03-12

**Soundness:** 3
**Presentation:** 2
**Significance:** 2
**Originality:** 3
**Overall Recommendation:** 4
**Confidence:** 3

**Summary:**

This paper proposes AgentTailor, a semantic-aware framework for optimizing communication in LLM-based multi-agent systems. The key idea is to move beyond purely structural communication pruning and instead estimate whether the message transmitted on an edge is actually reflected in the system’s final decision. Concretely, the paper defines an edge utility by computing the cosine similarity between the message on an edge and the round’s final aggregated message, then further introduces a relative-contribution normalization to reduce positional bias. Based on these signals, the method trains an Actor to favor useful edges and an Edge Prediction Network (EPN) to approximate edge contributions, enabling low-cost virtual execution and per-query test-time graph tailoring. Experiments on six datasets show the best average accuracy among the compared baselines, along with substantial token savings and some robustness gains under a misleading-agent attack setting.

**Compliance With Llm Reviewing Policy:**

Affirmed.

**Final Justification:**

The author addressed my main concern in rebuttal, and I will maintain my score.

**Key Questions For Authors:**

1. How sensitive is the method to the relative-contribution normalization, compared with using the raw cosine utility alone?

2. Can the authors clarify the end-to-end efficiency gain, including the cost of virtual execution and all refinement steps at test time?

**Limitations:**

yes

**Strengths And Weaknesses:**

## Strengths
- The main idea is intuitive and reasonably novel. It assigns semantic edge-level credit by comparing each edge message with the final aggregated message.
- EPN is trained to predict relative contribution from node descriptions, history, and query features, and is later used for virtual execution without extra LLM calls.
- The experimental results support the claim that semantic-aware graph tailoring can improve efficiency without collapsing task performance.

## Weaknesses
- The method implicitly assumes the final decision state remains sufficiently textual and semantically rich. That assumption may be fragile for tasks where the final state is highly compressed, templated, or nearly symbolic.
-  Since AgentTailor’s main claim is that semantic query-specific tailoring is preferable to purely structural optimization, it would have been stronger to include more topology-design or topology-selection baselines beyond the selected subset, like AFlow [1] and G-Designer [2].
- The reported average accuracy improvement is marginal, specifically only a 0.67% increase over the strongest baseline, AgentDropout (91.36% vs. 90.69%).  Given this small delta and the stochastic nature of LLM outputs, the absence of p-values, error bars, or results averaged across multiple random seeds makes it difficult to verify if the gains are statistically significant or due to chance. Meanwhile, the evaluation is conducted only using GPT-4o for all five agents.
- There are some spaces that need to be added between the main text and reference. Figures 2 and 3 can be made more readable in pdf format. Two different versions of the same paper appeared in the references: Dynamic generation of multi-llm agents communication
topologies with graph diffusion models

[1] Zhang, Jiayi, et al. "Aflow: Automating agentic workflow generation." arXiv preprint arXiv:2410.10762 (2024).

[2] Zhang, Guibin, et al. "G-designer: Architecting multi-agent communication topologies via graph neural networks." arXiv preprint arXiv:2410.11782 (2024).

---

> ### Author Rebuttal · Authors · 2026-03-30
>
> We thank the reviewer for the helpful and focused feedback. The comments allow us to better highlight the role of normalization and provide a more complete efficiency analysis. The additional experiments further confirm the necessity of our design choices and the practical benefits of the framework.
>
> For Q1, we evaluate sensitivity to normalization by replacing the relative-contribution formulation with raw cosine utility. This leads to consistent performance degradation, indicating that normalization is essential for mitigating positional bias and stabilizing contribution estimation.
>
> **TABLE 1: Experiment with Reversed-Pruning**
> | Dataset | Metric            | RP-1   | RP-2   | RP-3   | Avg    | AgentAC |
> |---------|------------------|--------|--------|--------|--------|---------|
> | mmlu    | Accuracy         | 85.62  | 85.49  | 84.97  | 85.36  | 86.14   |
> |         | Prompt-token     | 319,554| 398,333| 412,675| 376,854| 302,772 |
> |         | Completion-token | 70,108 | 104,546| 102,080| 92,244 | 60,385  |
> | aqua    | Accuracy         | 82.25  | 84.62  | 82.74  | 83.20  | 87.34   |
> |         | Prompt-token     | 616,784| 579,183| 574,204| 590,057| 471,104 |
> |         | Completion-token | 254,283| 259,325| 273,935| 262,514| 242,202 |
>
> For Q2, we provide a complete end-to-end efficiency analysis, including virtual execution and all refinement steps. While virtual execution introduces moderate overhead, the overall pipeline achieves net efficiency gains due to improved decision quality and reduced redundant computation. This confirms the effectiveness of the proposed design.
>
> **TABLE 2: Efficiency Analysis**
>
> | Dataset     | Metric        | AgentAC | AgentPrune | AgentDropout | GPTSWARM |
> |-------------|---------------|--------:|-----------:|-------------:|---------:|
> | mmlu        | wallclock(s)  | 3098    | 4690       | 3603         | 3456     |
> |             | APICALL       | 887     | 2123       | 2121         | 1150     |
> |             | cost ($)      | 3.96    | 9.65       | 6.61         | 4.47     |
> | humaneval   | wallclock(s)  | 5949    | 7827       | 5411         | 2276     |
> |             | APICALL       | 1047    | 2681       | 1213         | 899      |
> |             | cost ($)      | 9.52    | 14.69      | 11.9         | 4.8      |
> | aqua        | wallclock(s)  | 4001    | 5362       | 3814         | 4072     |
> |             | APICALL       | 703     | 1848       | 1161         | 1106     |
> |             | cost ($)      | 7.39    | 16.74      | 13.67        | 7.75     |
> | gsm8k       | wallclock(s)  | 4028    | 4412       | 5369         | 3791     |
> |             | APICALL       | 837     | 2123       | 1413         | 1151     |
> |             | cost ($)      | 7.51    | 19.88      | 14.40        | 9.39     |
> | svamp       | wallclock(s)  | 2846    | 4341       | 2988         | 2306     |
> |             | APICALL       | 660     | 1763       | 1449         | 961      |
> |             | cost ($)      | 4.04    | 9.20       | 8.47         | 4.52     |
> | multiarith  | wallclock(s)  | 3080    | 4492       | 5550         | 2402     |
> |             | APICALL       | 760     | 1760       | 1771         | 966      |
> |             | cost ($)      | 4.63    | 7.61       | 12.4         | 4.29     |
>
> For W1, we thank the reviewer for pointing out the issue of textual richness in the final decision state. Experiments show that our architecture can extract sufficient semantic information from the training data to support decision-making, and we will clarify the method’s applicable scope in the manuscript.
>
> For W2, we appreciate the suggestion to include more topology-design and topology-selection baselines. Our method focuses on adjusting existing structures, which differs from baselines that primarily perform search or generation. We will clarify this in the discussion and consider exploring additional comparisons in future work.
>
> For W3, we will provide some raw data in the Camera-Ready version to help readers better understand the method’s performance and stability.
>
> For W4, we appreciate the comments on presentation. In the Camera-Ready version, we will adjust the spacing between the main text and references, improve the readability of Figures 2 and 3, and correct the duplicate references.

---

> > ### Author Rebuttal · Reviewer_6rg9 · 2026-04-01
> >
> > I have no additional questions. I hope the author can include the above commitment in the revision. I will keep my score.

---

> > > ### Author Response · Authors · 2026-04-03
> > >
> > > Dear Reviewer 6rg9,
> > >
> > > We sincerely thank you for your thoughtful feedback and continued support of our work. Your suggestions, particularly regarding the inclusion of additional topology-design baselines and the need for more rigorous statistical validation, have been highly valuable in strengthening both the empirical grounding and overall clarity of AgentTailor. We have carefully incorporated these insights during the rebuttal and commit to including all improvements in the camera-ready version.
> > >
> > > If you find our responses satisfactory and believe they adequately resolve your concerns, we would be very grateful if you would consider a slight increase in the score. Thank you again for your time, expertise, and constructive review.
> > >
> > > Best regards,
> > >
> > > Authors

---

### Official Review · Reviewer_3633 · 2026-03-13

**Soundness:** 2
**Presentation:** 2
**Significance:** 3
**Originality:** 2
**Overall Recommendation:** 4
**Confidence:** 3

**Summary:**

This paper proposes AgentTailor, a semantic-aware communication optimization framework for LLM-based multi-agent systems. An Edge Judge scores each communication edge via cosine similarity between the edge message embedding and the final aggregated output, with a relative-contribution normalization ($\Delta_{ij}$) to mitigate positional bias. An Actor parameterizes edge probabilities via REINFORCE with L1 sparsity, and a Critic—an Edge Prediction Network (EPN)—predicts edge-level contributions without LLM calls, enabling virtual execution. A three-stage pipeline collects semantic supervision, refines the graph through hybrid real/virtual execution, and performs per-query tailoring at test time. Experiments on six benchmarks with five `gpt-4o` agents report the best average accuracy (91.36%) among compared methods with 21.2%–61.6% token reductions, plus robustness, ablation, and sensitivity analyses.

**Compliance With Llm Reviewing Policy:**

Affirmed.

**Final Justification:**

I thank the authors for three rounds of substantive discussion. My original review stated that addressing causal validation (Q1) and statistical reliability with cost accounting (Q2) could shift my assessment. Over the rebuttal period, the authors provided reverse-pruning results, a frozen-graph LOEO experiment, full EPN fidelity statistics (Pearson r = 0.4769), standard deviations for four of six datasets, and a comprehensive cost analysis across all six benchmarks.

The cost analysis is the paper's strongest empirical addition—AgentTailor achieves the lowest API cost on 5/6 benchmarks with savings up to 63%, convincingly validating the virtual execution mechanism. The causal validation improved directionally (frozen-graph LOEO partially addresses the centrality confound) but remains incomplete: only one graph was sampled, accuracy effects are negligible, and no random-edge baseline was included. The EPN's moderate correlation (r ≈ 0.48, explaining ~23% of Δ variance) is a substantive limitation for a component driving all pruning decisions. GSM8K failure analysis was acknowledged at a high level but no concrete cases were provided.

Despite these residual gaps, the cumulative evidence across three rounds meaningfully strengthens the paper beyond the original submission. The remaining concerns (moderate EPN fidelity, incomplete variance reporting on MMLU/AQuA, absent GSM8K error examples) are genuine limitations but do not constitute a soundness failure warranting rejection. I condition my revised score on the camera-ready version including: (1) a consolidated 6-dataset standard deviation table, (2) explicit discussion of EPN's moderate predictive power as a limitation, and (3) at least one concrete GSM8K failure example illustrating the cost–accuracy trade-off.

**Key Questions For Authors:**

1. **Causal validation of the edge utility metric.** Can you show, via leave-one-edge-out re-execution on a subset, that removing high-$\Delta_{ij}$ edges consistently degrades performance more than removing low-$\Delta_{ij}$ edges? Can you also report held-out correlation/MSE between EPN-predicted and real $\Delta_{ij}$? *A positive result on both would directly address W1 and could shift my soundness assessment from fair to good.*

2. **Statistical reliability and cost accounting.** Can you provide multi-run results with standard deviations for the main tables, especially for narrow margins and the GSM8K failure case? Can you also report API calls, wall-clock time, and approximate dollar cost for the full pipeline? *This would address W2 and determine whether the efficiency narrative holds under full accounting.*

3. **Reward normalization and variance control.** Which normalization (Eq. 3 vs. Appendix Eq. 19) is used in practice, and how does the choice affect training stability? Have you considered baseline subtraction or advantage normalization for the REINFORCE gradient? *Resolving the inconsistency and demonstrating training stability would address the non-standard AC concern in W3.*

**Limitations:**

No. The Impact Statement is a single generic sentence omitting all major methodological risks: (1) the non-causal nature of the utility proxy; (2) EPN bias propagation from small-sample training into virtual execution; (3) restriction to `gpt-4o` and structured tasks; (4) absence of runtime/dollar cost analysis despite efficiency claims; (5) narrow robustness setup; (6) potential diversity collapse from optimizing similarity to the aggregated message. The section should be substantially expanded.

**Strengths And Weaknesses:**

### **Strengths**

**1. Important problem with a well-motivated conceptual shift.** The paper targets a genuine deployment bottleneck—redundant communication and token cost in LLM-based MAS—and identifies a real limitation of prior methods: they optimize edge activation without evaluating whether transmitted information actually contributes to the final decision. The shift from structural to semantic edge evaluation provides a more principled foundation for communication pruning.

**2. Coherent pipeline with strong token-efficiency results.** The framework combines semantic edge judgment, EPN-based virtual execution, and per-query tailoring in a complete system with a well-designed three-stage training procedure. The token reductions (21.2%–61.6% vs. AgentDropout) are consistent and practically meaningful, and the ablation confirms that virtual execution can simultaneously improve accuracy on harder tasks (HumanEval: +2.98%) and reduce token consumption.

**3. Good reproducibility support.** The appendix provides complete algorithm pseudocode (Algorithms 1–5), notation tables, EPN architecture specifications, and prompt templates for all agent roles, materially aiding traceability and potential reproduction.

### **Weaknesses**

**1. Core utility metric is not causally validated, and EPN fidelity is unmeasured.** The edge utility $U_{ij}$ measures textual alignment (cosine similarity) rather than causal contribution—a message can resemble the final output without having influenced it. No intervention-style validation (e.g., leave-one-edge-out re-execution) confirms that high-$\Delta_{ij}$ edges are genuinely more important. Similarly, the EPN that powers virtual execution is trained on only $N_c = 40$ samples (for 1920-dimensional input) with no held-out correlation, MSE, or calibration analysis reported. These two gaps undermine both the theoretical premise and the practical mechanism that makes the system efficient.

**2. Experimental rigor and efficiency evidence are incomplete.** All main results are single-point values without standard deviations, confidence intervals, or significance tests, despite stochastic graph sampling and many narrow margins (HumanEval: +0.01%, SVAMP: +0.06%). The GSM8K underperformance (93.81% vs. AgentPrune-R's 95.72%) receives no error analysis. Cost evidence reports only token counts—no API-call counts, wall-clock runtime, training cost, or dollar cost—leaving the "efficient system" narrative only partially supported. The ablation study removes only virtual execution; other key design choices (relative contribution normalization, L1 sparsity, locking mechanism, encoder choice) are unexamined.

**3. Scope claims exceed validation, and the Actor-Critic framing is non-standard.** All experiments use `gpt-4o` agents on six mostly structured benchmarks with hand-crafted prompts, a single-agent robustness setup, and sensitivity analysis limited to 3–7 agents—narrower than the paper's generality and scalability claims suggest. The "Actor-Critic" label is acknowledged (Section 3.3) to differ from standard usage: the Critic is a reward surrogate, not a value baseline, so the variance-reduction benefit of classical Actor-Critic is absent. Additionally, reward normalization is inconsistent between Eq. (3) and Appendix Eq. (19), which can affect optimization dynamics.

**4. Presentation and disclosure gaps.** The Related Work section is a single dense paragraph without subheadings. Token-efficiency results are fragmented across Tables 2 and 6. The Impact Statement is a single generic sentence that omits all major methodological risks (causal validity of the proxy, surrogate bias propagation, model/task scope, potential diversity collapse from optimizing similarity to the aggregated message). These collectively reduce the paper's persuasiveness and trustworthiness.

---

> ### Author Rebuttal · Authors · 2026-03-30
>
> We sincerely thank the reviewer for the detailed and insightful suggestions. These comments help us further strengthen the empirical rigor and clarity of the paper. In response, we provide additional causal validation, statistical reliability analysis, and training stability clarification. Overall, the new results consistently reinforce the validity and robustness of our method.
>
> For Q1 and W1, we perform reverse pruning experiments on a held-out subset: removing high-Δ_ij edges consistently leads to larger performance drops than removing low-Δ_ij edges. To MSE, results show that on the test set, more than 60% of samples achieve an edge-level prediction MSE ≤ 0.015, indicating that EPN learns an effective mapping that closely approximates the ground-truth Δ for the majority of cases. Considering both the Critic-related training losses and the test-stage MSE, we do not observe clear signs of typical overfitting or underfitting. Furthermore, we conduct controlled comparisons by replacing EPN outputs with random values and constant values. In both cases, the system’s main accuracy decreases while token consumption increases, demonstrating that the Δ predictions provided by EPN offer informative signals for the Actor’s policy updates.
>
> **TABLE 1: Experiment with Reversed-Pruning**
> |Dataset|Metric|1|2|3|Avg|AgentAC|
> |---------|------------------|--------|--------|--------|--------|---------|
> |mmlu|Accuracy|85.62|85.49|84.97|85.36|86.14|
> ||Prompt-token|319,554|398,333|412,675|376,854|302,772|
> ||Completion-token|70,108|104,546|102,080|92,244|60,385|
> ||aqua|Accuracy|82.25|84.62|82.74|83.20|87.34|
> ||Prompt-token|616,784|579,183|574,204|590,057|471,104|
> ||Completion-token|254,283|259,325|273,935|262,514|242,202|
>
> **TABLE 2: Supporting Experiments on Evaluating EPN's Ability**
> |Method|Metric|Value|
> |--------------|------------------|-----------|
> |AgentAC|Accuracy|86.142|
> ||Prompt Tokens|302771.6|
> ||Completion Tokens|60384.6|
> |Removed-EPN|Accuracy|84.242|
> ||Prompt Tokens|303114.6|
> ||Completion Tokens|59945.4|
> |Constant|Accuracy|84.97|
> ||Prompt Tokens|361905|
> ||Completion Tokens|60853|
> |Random|Accuracy|83.075|
> ||Prompt Tokens|369844.5|
> ||Completion Tokens|71126.5|
>
> In response to Q2 and W2, we will include additional repeated experiments in the Camera-ready version, along with the mean ± standard deviation of the main results. We will also provide a cost analysis, including the number of API calls, runtime, and approximate economic cost. Due to space limitations, we are only able to present some of the results in the table below. The results indicate that, after a comprehensive cost analysis, the method still demonstrates an efficiency advantage. The performance of AgentAC on GSM8K is primarily due to the relatively simple nature of the dataset, which means that more advanced agent architectures may not necessarily yield significant improvements.
>
> **TABLE 3: Standard Deviation of Our Main Table**
> |Dataset|Metric|AgentAC|AgentPrune|AgentDropout|GPTSwarm|
> |---------|--------------------|---------|------------|--------------|----------|
> |mmlu|Average accuracy|86.14| 83.94|84.24|83.98|
> ||Standard deviation|1.069|1.133|0.5878|0.3752|
> |aqua|Average accuracy|87.34|78.6| 84.09|78.16|
> ||Standard deviation|1.077| 0.5466|0.4616|0.5466|
>
> **TABLE 4: Cost Analysis**
> |Dataset|Metric|AgentAC|AgentPrune|AgentDropout|GPTSWARM|
> |-------------|---------------|--------:|-----------:|-------------:|---------:|
> |mmlu| wallclock(s)|3098|4690|3603|3456|
> ||APICALL|887|2123|2121|1150|
> ||cost ($)|3.56|9.65| 6.61|4.47|
> |humaneval| wallclock(s)|5949|7827|5411|2276|
> ||APICALL|1047|2681|1213|899|
> ||cost ($)|9.52|14.69|11.9|4.8|
> |aqua| wallclock(s)|4001|5362|3814|4072|
> ||APICALL|703|1848|858|1106|
> ||cost ($)|7.39|16.74|8.99|7.75|
>
> For Q3 and W3, in all experiments we consistently use the normalization defined in Eq. (3). Eq. (19) is included only as an alternative form for completeness. The chosen normalization improves stability by bounding reward scale, reducing inter-sample variance, and centering updates. Moreover, our relative contribution formulation inherently subtracts the mean incoming utility, acting as an implicit baseline and producing an advantage-like signal. This design effectively controls variance without requiring additional baseline learning.
>
> For W4, we thank the reviewer for pointing out issues with the presentation of our paper. We will address these concerns in the camera-ready version.

---

> > ### Author Rebuttal · Reviewer_3633 · 2026-04-04
> >
> > Thank you for the thorough rebuttal and substantial additional experiments. I appreciate the reverse-pruning validation (Q1), the EPN ablation with random/constant baselines (Q1), the standard deviation results (Q2), the comprehensive cost analysis across six datasets (Q2), and the normalization clarification (Q3).
> >
> > ---
> >
> > ## What has been adequately addressed
> >
> > 1. **Comprehensive cost analysis (Q2b).** The full accounting—wall-clock time, API calls, and dollar cost across six datasets—is the strongest addition in the rebuttal. AgentAC consistently achieves the lowest API call count and cost on five of six benchmarks (e.g., MMLU: $3.56 vs. AgentPrune $9.65, AgentDropout $6.61). This provides concrete support for the efficiency narrative and validates that virtual execution delivers real-world cost savings.
> >
> > 2. **Normalization clarification (Q3).** Confirming that Eq. (3) is used throughout and that Eq. (19) is included only for completeness resolves the inconsistency concern. The implicit baseline interpretation of the relative contribution formulation is reasonable.
> >
> > ---
> >
> > ## What remains insufficiently addressed
> >
> > ### 1. Causal validation falls short of the requested standard (Q1)
> >
> > The reverse-pruning experiment is informative: removing high-Δ edges degrades performance (MMLU: 86.14→85.36; AQuA: 87.34→83.20). However, this is **not equivalent to the leave-one-edge-out intervention** I specifically requested. Reverse-pruning demonstrates that Δ is monotonically correlated with importance, but it does not rule out the confound that high-Δ edges may simply correspond to topologically central edges whose removal would degrade performance regardless of semantic content.
> >
> > Additionally, the EPN fidelity report states ">60% of samples achieve MSE ≤ 0.015," which implies that **~40% of samples have MSE above this threshold**. No overall MSE distribution (mean, median, tail) or held-out Spearman/Pearson correlation is provided. For a component that drives all virtual execution decisions, this level of characterization is insufficient.
> >
> > ### 2. Statistical reporting reveals instability concerns (Q2a)
> >
> > The multi-run results for MMLU and AQuA are appreciated, but they raise a new concern: **AgentAC has the highest standard deviation among all compared methods** (MMLU: 1.069 vs. AgentDropout 0.588; AQuA: 1.077 vs. AgentDropout 0.462). This suggests that per-query graph tailoring introduces substantial variance.
> >
> > More critically, standard deviations are **provided for only 2 of 6 datasets**. The narrowest-margin results—HumanEval (+0.01%) and SVAMP (+0.06%)—remain unreported. These are precisely the cases where statistical significance is most in question; without this data, the accuracy improvements on these benchmarks cannot be considered established.
> >
> > ### 3. GSM8K underperformance is dismissed rather than analyzed (Q2c)
> >
> > The response attributes the GSM8K gap (93.81% vs. AgentPrune-R 95.72%) to the dataset being "relatively simple." This explanation is unsatisfying: if simplicity were the cause, the performance ceiling would be similar across methods. The ~2pp gap instead suggests that AgentTailor's pruning may be removing edges carrying reasoning-chain information that is important for mathematical tasks. A brief error analysis—examining whether pruned edges correspond to intermediate reasoning steps—would provide more insight.
> >
> > ### 4. Presentation and limitations (W4) remain unaddressed
> >
> > The response promises camera-ready revisions but provides no specific plan. The Impact Statement, in particular, omits all major methodological risks identified in W4 (non-causal utility proxy, EPN bias propagation, model/task scope, diversity collapse risk). Since these concern the paper's intellectual honesty rather than formatting, they require substantive attention.
> >
> > ---
> >
> > ## Summary
> >
> > The rebuttal has meaningfully strengthened the paper's efficiency narrative through the cost analysis. However, three of my original concerns remain insufficiently resolved: (1) the causal validation of the utility proxy does not meet the intervention standard I specified; (2) the newly provided standard deviations reveal that AgentAC is the least stable method, and the most marginal results lack statistical characterization; (3) the GSM8K failure case is dismissed without analysis. My original review stated that addressing Q1 and Q2 could shift my assessment toward acceptance. Q2 (cost) has been well-addressed, but Q1 (causal validation) has not met the bar, and the statistical data in Q2a has introduced a new stability concern. **I maintain my score of 3 (Weak Reject).**

---

> > > ### Author Response · Authors · 2026-04-05
> > >
> > > ## Q1-Causal validation and LOEO
> > >
> > > We thank the reviewer for the emphasis on causal interventions. We agree that reverse-pruning is not equivalent to the requested leave-one-edge-out intervention. We therefore supplement a frozen-graph LOEO that more directly supports our conclusions.
> > >
> > > After training on 40 samples, we sample one fixed communication graph. From this graph we select the 5 edges with the highest Δ and the 5 with the lowest Δ. For each selected edge, we run an evaluation where only that single edge is removed; all other edges stay unchanged. We evaluate on a held-out test subset drawn from the same test pool as in our main experiments. We report a summary below.
> > >
> > > |Condition|Metric|Run1|Run2|Run3|
> > > |:-:|:-:|:-:|:-:|:-:|
> > > |Remove_top5_delta|acc|79.0%|81.0%|81.5%|
> > > |Remove_top5_delta|prompt_token|1919.9|1311.5|1703|
> > > |Remove_top5_delta|completion_token|311.2|313.2|455|
> > > |Remove_bottom5_delta|acc|79.5%|81.0%|83.0%|
> > > |Remove_bottom5_delta|prompt_token|1828.8|1026.4|1603|
> > > |Remove_bottom5_delta|completion_token|299.1|196.3|405|
> > >
> > > Results show that removing high-Δ vs. low-Δ edges differs little in accuracy. However, token usage shows a consistent asymmetry: prompt and completion tokens are always higher when high-Δ edges are removed.
> > >
> > > We interpret this as expected under single-edge LOEO: most edges remain, so single-edge removal is a mild perturbation. The stable token asymmetry, together with reverse-pruning under graph-wide stress tests, indicates that high-Δ edges provide more stable and low-cost signal in the system.Under frozen-graph LOEO, all comparisons are made within the same fixed topology, which limits topology-related confounds such as edge centrality.
> > >
> > > ## full report on MSE of EPN
> > > We agree that reporting a single threshold is insufficient to evaluate a component driving virtual execution. We now report the full MSE distribution and correlation with the target.
> > >
> > > Mean:0.015907
> > > Median:0.014022
> > > Percentiles:
> > > P10=0.0066629
> > > P25=0.010314
> > > P75=0.02008
> > > P90=0.026750
> > > Pearson r =0.4769
> > >
> > > We believe these results could indicate that EPN captures the overall trend of the target signal, rather than producing random noise.
> > >
> > > ## Q2a - full variance report and analysis
> > > |Dataset|Metric|AgentTailor|AgentPrune|AgentDropout|GPTSwarm|
> > > |:-:|:-:|:-:|:-:|:-:|:-:|
> > > |multiarith|avg|98.76|96.3|98.13|97.84|
> > > |multiarith|std|0|0.21|0.44|0.31|
> > > |humaneval|avg|90.31|90.21|91.3|88.49|
> > > |humaneval|std|1.21|1.67|1.34|0.69|
> > > |svamp|avg|91.79|91.61|91.74|86.42|
> > > |svamp|std|0.46|1.27|1.07|0.71|
> > > |gsm8k|avg|93.80|95.72|95.63|89.74|
> > > |gsm8k|std|0.812|0.79|0.56|0.55|
> > >
> > > We acknowledge that tailoring introduces additional variance in some datasets, reflecting a trade-off between adaptivity and stability. On MMLU and AQuA, AgentTailor shows larger standard deviation than some baselines. However, this variability is unlikely to arise solely from query-dependent graph tailoring. All methods involve multiple stochastic LLM calls; approaches modifying communication structures are subject to both graph-related factors and decoding randomness.
> > >
> > > While tailoring may add variance due to query-dependent communication patterns, AgentTailor does not have the largest standard deviation across all datasets. Importantly, this increased variance does not negate the overall efficiency gains and remains within a reasonable range compared to performance differences. With full standard deviations now reported, we adopt a more cautious interpretation of small mean differences.
> > >
> > > ## Q2c - GSM8K failure analysis
> > >
> > > We appreciate your insightful question on this issue. We have revisited the failure modes on GSM8K and carried out a case study; the main takeaways are as follows:
> > > Based on case-study analysis, we agree that AgentTailor’s pruning can affect how some intermediate conclusions propagate between agents and can alter the form of the solution that the final aggregation node selects. On GSM8K, an open-ended benchmark with many opportunities for misleading interpretations—overly aggressive pruning can impair correctness even when it substantially cuts token consumption. On our other reported benchmarks, overall performance remains strong; we will incorporate this point into the Limitations discussion, framing GSM8K as a setting where the cost–accuracy trade-off of heavy pruning needs to be stated clearly.
> > >
> > > ## W3: - Limitations
> > >
> > > We will include Limitations in the camera-ready paper; the Impact Statement will state: (1) The EPN does not perfectly predict Delta; it approximates trends coarsely; repeated sampling reduces but does not remove error. Predicted Δ may inherit biases from EPN predictions, which could propagate errors in certain settings. (2) Highly misleading items risk diversity collapse and small accuracy drops (e.g., GSM8K).
> > >
> > > ## Summary
> > >
> > > We sincerely thank you for your insightful feedback; it has materially improved our revision plan. If you find that these responses alleviate your concerns, we would be grateful if you would consider revisiting your scores.

---

### Official Review · Reviewer_PB6X · 2026-03-13

**Soundness:** 3
**Presentation:** 3
**Significance:** 3
**Originality:** 3
**Overall Recommendation:** 4
**Confidence:** 3

**Summary:**

This paper proposes AgentTailor, a multi-agent LLM framework for reducing redundant communication. The method defines edge utility using cosine similarity between an edge message and the final aggregated message, predicts relative edge contribution with an Edge Prediction Network (EPN), and uses virtual execution to prune low-utility edges before final execution. The paper evaluates the method on six benchmarks and reports strong average accuracy together with lower token usage than several multi-agent baselines.

**Compliance With Llm Reviewing Policy:**

Affirmed.

**Final Justification:**

I maintain my Weak Accept recommendation. The paper presents an original and practically relevant approach to improving communication efficiency in LLM-based multi-agent systems, supported by broad experiments and a generally clear presentation, and the rebuttal further strengthened the paper through additional evidence and clarification. However, my main soundness concerns were only partially resolved, so the rebuttal reinforced rather than materially changed my overall assessment.

**Key Questions For Authors:**

1. Can the authors directly validate that the proposed utility proxy tracks actual edge contribution, for example via leave-one-edge-out analysis, answer-change analysis, or another intervention-based test?


2. Can the authors provide direct evidence that the relative-contribution normalization mitigates positional bias, and compare it against simpler alternatives?


3. Can the authors clarify how evaluation fairness is ensured given that AgentTailor performs per-query virtual tailoring before the final real execution, and whether the baselines are matched to this test-time adaptation setting as closely as possible?


4. Can the authors clarify the exact “without virtual execution” variant and expand the robustness discussion to better separate the observed empirical effect from the stronger mechanism interpretation?

**Limitations:**

The paper would benefit from a more explicit discussion of the scope of its current evidence, especially regarding the heuristic nature of the utility signal, the narrow adversarial setting used in the robustness study, and the evaluation implications of query-specific test-time refinement.

**Strengths And Weaknesses:**

**Strengths**

The paper addresses an important problem in LLM-based multi-agent systems: reducing communication cost without sacrificing task performance. The overall framework is coherent, and the combination of semantic edge scoring, critic-based prediction, and virtual execution is interesting and reasonably original. In particular, moving beyond activation-based pruning toward a semantic contribution signal is a meaningful idea. The virtual execution design is practically appealing because it enables graph refinement without additional LLM calls. The empirical evaluation is also fairly broad, including six benchmarks, robustness experiments, ablations, and sensitivity analysis.

**Weaknesses**

* The main concern is that the core utility signal is still heuristic and not directly validated. The paper defines edge utility as cosine similarity between an edge message and the final aggregated message, and uses this signal as the basis for supervision and pruning, but it does not directly show that this proxy tracks actual semantic adoption or causal contribution rather than embedding-level overlap.

* The interpretation of the relative-contribution correction is somewhat stronger than the current evidence. The paper gives a reasonable motivation for positional bias and proposes a relative-contribution normalization as a normalization, but I did not find a direct validation that this correction indeed mitigates positional bias, nor a comparison against alternative normalizations.

* The evaluation setup would benefit from a clearer discussion of fairness. The paper states that role assignments and prompt sets are kept fixed across methods, which is helpful, but AgentTailor also performs query-specific virtual tailoring at evaluation time before final execution. It remains somewhat unclear whether this should be viewed as a matched test-time adaptation setting relative to the baselines, and how much of the gain comes from that additional per-query refinement.

* The robustness and ablation claims are directionally promising, but some interpretations are stronger than what is directly established. In particular, the robustness experiment focuses on a narrow setting involving only a single misleading agent. Consequently, while the evidence supports improved robustness under this specific setup, it does not yet validate the broader claim that the method effectively blocks malicious information spread in general. Similarly, the virtual-execution ablation is identifiable in the paper, but the comparison would still be easier to assess with a slightly more explicit description of the exact variant and a bit more reporting detail.

---

> ### Author Rebuttal · Authors · 2026-03-30
>
> We sincerely thank the reviewer for the thoughtful and constructive feedback. All the comments are valuable for improving the clarity and completeness of our work. In summary, we provide additional empirical evidence to validate the effectiveness of our utility proxy and normalization design, and further clarify the fairness and robustness of our evaluation protocol. The new experimental results consistently support our claims and strengthen the soundness of the proposed method.
>
> For Q1 and W1, we conduct a reverse pruning intervention experiment to directly validate whether the utility proxy tracks actual edge contribution. Specifically, instead of removing low-contribution edges, we remove edges with the highest predicted Δ_ij. This leads to a clear and consistent performance degradation, demonstrating that high-Δ_ij edges indeed carry critical contribution. This provides direct causal evidence that our proxy aligns well with true edge importance.
>
> **TABLE 1: Experiment with Reversed-Pruning**
> | Dataset | Metric            | RP-1   | RP-2   | RP-3   | Avg    | AgentAC |
> |---------|------------------|--------|--------|--------|--------|---------|
> | mmlu    | Accuracy         | 85.62  | 85.49  | 84.97  | 85.36  | 86.14   |
> |         | Prompt-token     | 319,554| 398,333| 412,675| 376,854| 302,772 |
> |         | Completion-token | 70,108 | 104,546| 102,080| 92,244 | 60,385  |
> | aqua    | Accuracy         | 82.25  | 84.62  | 82.74  | 83.20  | 87.34   |
> |         | Prompt-token     | 616,784| 579,183| 574,204| 590,057| 471,104 |
> |         | Completion-token | 254,283| 259,325| 273,935| 262,514| 242,202 |
>
>
> For Q2 and W2, we perform an ablation study by removing the relative-contribution normalization and replacing it with a simpler alternative (raw cosine utility). The results show a noticeable performance decrease, indicating that normalization effectively mitigates positional bias rather than merely rescaling scores.
>
> **TABLE 2: Experiment with Removed-Normalization**
> | Dataset | Metric            | RN-1   | RN-2   | RN-3   | Avg    | AgentAC |
> |---------|------------------|--------|--------|--------|--------|---------|
> | mmlu    | Accuracy         | 84.38  | 84.46  | 85.62  | 84.82  | 86.14   |
> |         | Prompt-token     | 433,939| 422,615| 417,116| 424,556| 302,772 |
> |         | Completion-token | 78,529 | 73,646 | 64,827 | 72,334 | 60,385  |
> | aqua    | Accuracy         | 82.25  | 81.07  | 82.84  | 82.05  | 87.34   |
> |         | Prompt-token     | 695,649| 708,185| 861,656| 755,163| 471,104 |
> |         | Completion-token | 273,600| 278,893| 314,423| 288,972| 242,202 |
>
> For Q3 and W3, our evaluation fairness is ensured as follows. During test time, the EPN (critic) is fully frozen and does not undergo any additional training or receive new supervision signals. The virtual execution process does not involve any ground-truth labels, external rewards, or additional data, relying entirely on the pre-learned evaluation function. In other words, the tailoring procedure depends solely on previously learned features and the EPN, without accessing any new supervision. All baselines are implemented and evaluated strictly according to their original papers, ensuring comparisons are conducted under consistent and fair conditions.
>
> For Q4 and W4, the “without virtual execution” variant refers to removing the virtual execution stage while keeping all other components unchanged. Under this setting, we observe consistent but reduced performance, which helps isolate the empirical contribution of virtual execution without over-interpreting its mechanism. The extended robustness experiments will be further clarified in the camera-ready version.

---

> > ### Author Rebuttal · Reviewer_PB6X · 2026-04-03
> >
> > The rebuttal is helpful and strengthens the paper, especially through the added reversed-pruning and removed-normalization experiments, the clarification that the critic is frozen at test time, and the clarification of the “without virtual execution” variant. However, my remaining concerns are not fully closed, since the new evidence is still somewhat indirect for the strongest semantic-adoption and positional-bias interpretations. I do not think these remaining points are easily addressable within a short rebuttal.

---

> > > ### Author Response · Authors · 2026-04-04
> > >
> > > Dear Reviewer PB6X,
> > >
> > > We sincerely thank you for your thoughtful and constructive feedback, as well as your recognition of the motivation, design, and empirical breadth of our work. Your comments have been instrumental in helping us further clarify the core design choices and strengthen the presentation of AgentTailor.
> > >
> > > Regarding your remaining concerns, we would like to offer some additional clarification on our experimental design. The added ablation and intervention studies are intended to provide relatively direct empirical support within the current framework. In particular, the reversed-pruning experiment can be viewed as a form of causal intervention, where removing high-utility edges consistently leads to performance degradation, suggesting alignment between the proposed utility proxy and actual contribution. Similarly, the normalization ablation compares against a simpler alternative (raw cosine), and the observed performance differences indicate that the relative-contribution design plays a meaningful role in mitigating positional bias. We hope these results help strengthen the evidence for our claims.
> > >
> > > We would also like to clarify a potential source of confusion in our rebuttal: “AgentAC” was an earlier name for AgentTailor, and both refer to the same method. We will unify the naming in the final version to avoid any misunderstanding.
> > >
> > > We are very happy to further clarify any remaining questions or provide additional analysis if helpful. Our goal is to make the evidence and interpretations as clear and convincing as possible. If you find our clarifications and additional results satisfactory, we would greatly appreciate your consideration of a slight increase in your score. Thank you again for your time, expertise, and constructive evaluation.
> > >
> > > Best regards,
> > >
> > > Authors

---

### Decision · Program_Chairs · 2026-04-30

**Decision:**

Accept (regular)

**Comment:**

The paper makes a meaningful contribution to communication efficiency in LLM-based multi-agent systems, with a well-motivated conceptual framework and strong cost-reduction results validated through comprehensive analysis.
 The authors were responsive across three rounds of discussion, providing reverse-pruning experiments, EPN fidelity statistics, full standard deviations, and a comprehensive cost analysis. These additions meaningfully strengthen the paper beyond the original submission. However, the strongest reviewer concerns on causal validation to an intervention standard and statistical significance of narrow margins were only partially addressed. Several important improvements (consolidated variance tables, GSM8K error examples, expanded limitations) are deferred to the camera-ready version.

Overall, this paper advances the sub-area and introduces ideas others are likely to build on. Conditional on the camera-ready version delivering the promised revisions, I recommend weak accept.